# GPLQ: A General, Practical, and Lightning QAT Method for Vision Transformers

**Guang Liang**[1,2,4] **Xinyao Liu**[3] **Jianxin Wu**[1,2*]

[1]National Key Laboratory for Novel Software Technology, Nanjing University, China
[2]School of Artificial Intelligence, Nanjing University, China
[3]University of Science and Technology of China, Hefei, China
[4]Zhongguancun Academy, Beijing, China

`liangg@lamda.nju.edu.cn, liuxinyao@mail.ustc.edu.cn, wujx2001@nju.edu.cn`

## Abstract

Vision Transformers (ViTs) are essential in computer vision but are computationally intensive, too. Model quantization, particularly to low bit-widths like 4-bit, aims to alleviate this difficulty, yet existing Post-Training Quantization (PTQ) and Quantization-Aware Training (QAT) methods exhibit significant limitations. PTQ often incurs substantial accuracy drop, while QAT achieves high accuracy but suffers from prohibitive computational costs, limited generalization to downstream tasks, training instability, and lack of open-source codebase. To address these challenges, this paper introduces General, Practical, and Lightning Quantization (GPLQ), a novel framework designed for efficient and effective ViT quantization. GPLQ is founded on two key empirical insights: the paramount importance of activation quantization and the necessity of preserving the model's original optimization "basin" to maintain generalization. Consequently, GPLQ employs a sequential "activation-first, weights-later" strategy. Stage 1 keeps weights in FP32 while quantizing activations with a feature mimicking loss in only 1 epoch to keep it in the same "basin", thereby preserving generalization. Stage 2 quantizes weights using a PTQ method. As a result, GPLQ is 100x faster than existing QAT methods, lowers memory footprint to levels even below FP32 training, and achieves 4-bit model performance that is highly competitive with FP32 models in terms of both accuracy on ImageNet and generalization to diverse downstream tasks, including fine-grained visual classification and object detection. We release an easy-to-use open-source toolkit supporting multiple vision tasks at `https://github.com/wujx2001/GPLQ`.

## 1 Introduction

Vision Transformer (ViT) [6, 34] has emerged as the mainstream backbone network in computer vision, but it demands substantial computational and memory resources. Model quantization is one of the key techniques to address this challenge by reducing the numerical precision of model parameters and/or activation values[18, 19]. However, existing quantization methods still faces challenges, especially in low-bit (e.g., 4-bit) quantization.

Mainstream methods include Post-Training Quantization (PTQ) [25] and Quantization-Aware Training (QAT) [7]. PTQ has fast speed and low resource consumption, but often leads to large accuracy drop under 4-bit quantization [20]. On the other hand, QAT simulates quantization operations during training and enables higher accuracy than that of PTQ, or *even higher than that of floating-point models*. Nevertheless, in this paper we will show that existing QAT methods have inherent limitations:

---

*Corresponding author.

39th Conference on Neural Information Processing Systems (NeurIPS 2025).

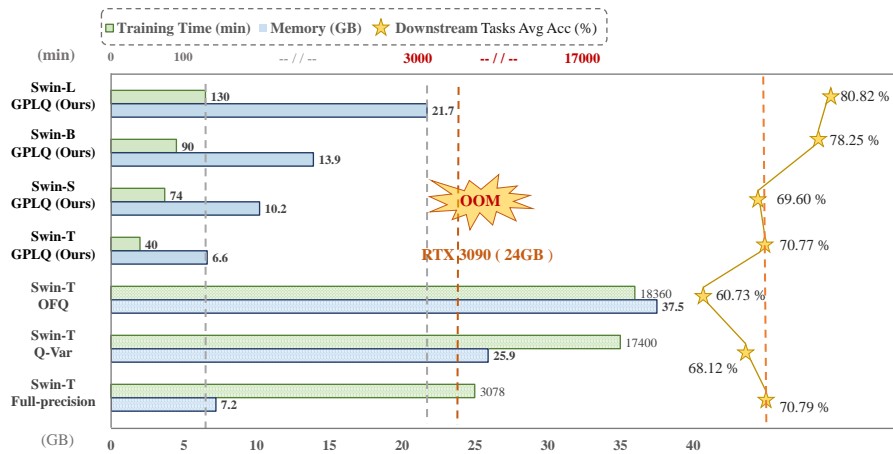

Figure 1: Core advantages of our GPLQ: Generality, Practicality, and Lightning efficiency.

- **High Computational Costs.** QAT requires lengthy fine-tuning of the entire model. Training time and GPU memory required in QAT often *far exceed those for training the FP32 model* [18]. This makes QAT *cumbersome and very slow* for deployment in real-world applications.

- **Limited Generalization Ability.** QAT methods often boast higher accuracy than their FP32 counterparts. However, in this paper we will show that such models are *generalizing worse* than FP32 or PTQ quantized models in downstream tasks. That is, they are likely *non-generalizable* beyond ImageNet [4], the dataset on which they were trained.

- **Training Instability and Complexity.** QAT is prone to training instability [15], and complex Knowledge Distillation (KD) techniques [19, 15] severely increase memory footprint. Some also rely on external, extremely powerful teacher models, which are *not* available in practical scenarios. In short, existing QAT methods are *not practical*.

- **Classification Only and Code Missing.** Open-source code for QAT is rare, and is only for classification when it exists. This further makes QAT *impractical* for real-world applications.

To this end, we propose GPLQ (General, Practical, and Lightning Quantization). The core objective of GPLQ is to provide a quantization solution that is far more training-efficient than traditional QAT, superior to PTQ in accuracy and generalization, easy to use, and highly practical. As a result, Figure 1 demonstrates 3 core advantages of GPLQ.

- **General.** GPLQ exhibits excellent average accuracy on multiple *downstream tasks*: close to or even surpassing FP32 models, and significantly outperforming existing QAT methods.

- **Practical.** GPLQ has *very small training memory footprint* (far lower than existing QAT methods), which avoids out-of-memory (OOM) issues in many applications and enables quantization of larger models. GPLQ's design allows it to be conveniently applied to *other tasks such as object detection*.

- **Lightning.** GPLQ is blazingly fast: *hundreds of times faster* than existing QAT methods.

GPLQ is based on our empirical findings. First, *activations are far more important than weights* in low-bit quantization. Second, quantization should not change its optimization "basin" (i.e., *avoid jumping out of the current local minimum*) in order to keep the generalization ability.

Based on these findings, GPLQ adopts a sequential quantization paradigm. First, activations are quantized with weights kept at FP32. To maintain generalization, we draw inspiration from TCS [40] and employ a PCA-based feature mimicking loss to guide the quantized model's feature outputs to approximate those of the original FP32 model (i.e., stay in the same basin). Second, after activations are quantized, existing efficient PTQ methods are used to quantize the weights. This "activation-first, weights later" strategy not only drastically reduces QAT *training time from days to 1-2 hours* and

with *memory footprint even lower than FP32 training*, but also allows a 4-bit model to achieve both accuracy and generalization nearly identical to the original FP32 model. The main contributions are:

1. **Insights.** We reveal that activation quantization is the main bottleneck in QAT, and staying in the original optimization basin is crucial for generalization.

2. **GPLQ.** We propose "activation-first" sequential quantization: first optimize activations then quantize weights via PTQ.

3. **Code.** GPLQ provides an easy-to-use quantization tool supporting classification, detection and other downstream tasks. We will open-source GPLQ upon paper acceptance.

## 2 Related Work

Model quantization aims to enhance model efficiency by reducing the numerical precision of weights and activations in neural networks [29].

**Post-Training Quantization (PTQ)**. PTQ operates without retraining, requires only a small calibration set, and is very fast. Various techniques have been proposed: AIQViT [16], GPTQ [8], PTQ4ViT [39]), SmoothQuant [36], AWQ [21]. More methods like RepQ-ViT [20] and QwT [9] perform optimization through scale reparameterization and lightweight compensation modules, respectively. Accuracy degradation remains a severe challenge in low-bit scenarios.

The second stage of GPLQ employs PTQ to quantize weights. Since activations have been quantized via QAT in the first stage, PTQ's duty changes from W32A32 → W4A4 to W32A4 → W4A4.

**Quantization-Aware Training (QAT)**. QAT introduces simulated quantization during training or fine-tuning, and achieves higher accuracy than PTQ. It often uses a Straight-Through Estimator (STE) to handle gradient [7]. Research directions include learning quantization scales [7], improving training stability) (OFQ [23], Quantization-Variation [15]), enhancing efficiency (EfficientQAT [3]), and specific optimizations for ViTs (e.g., Q-ViT [19], PackQViT [5]).

The bottlenecks of QAT are high computational cost, training instability, and potential degradation in generalization ability. GPLQ, with an extremely short QAT stage (*only 1 epoch*) focused solely on activations, effectively alleviates the cost, stability, and generalization issues of traditional QAT.

**Knowledge Distillation (KD) in QAT**. KD [14, 11, 35, 41] is often used in QAT to learn from the FP32 model or even a much stronger, external teacher. Researchers have proposed methods (DGD [39], MCKD [15]) that are both heavy and complex. TCS [40] offers an efficient approach by capturing the linear subspace of the teacher model's features through Principal Component Analysis (PCA) for knowledge transfer. GPLQ draws inspiration from TCS, aiming at efficiently transferring knowledge to maintain generalization and avoiding the high costs of complex distillation.

## 3 Methodology

GPLQ is directly derived from two empirical insights in the ViT quantization process.

### 3.1 Activations are Crucial & Stay in the Same Basin

Our first empirical insight is that *activations are more critical than weights* in quantization. Taking a ViT network such as DeiT [32] pre-trained on ImageNet-1k [4], we independently applied 4-bit PTQ (using a percentile-based per-channel quantization calibration method) to either weights (with activations kept at FP32) or activations (with weights kept at FP32). Figure 2 shows the results, which consistently indicates that quantizing activations to 4 bits (weights at FP32) leads to larger Top-1 accuracy drop than quantizing weights to 4 bits (activations at FP32). That is, activations face more severe challenges under low-bit quantization compared to weights. This finding leads to our "activation-first" quantization strategy in GPLQ.

QAT methods heavily adjust their weights, and make them stay in dramatically different basins (local minima) in the loss landscape before and after QAT learning. That is, they are significantly different from the initial FP32 model. Although they achieve high accuracy on the pre-training ImageNet data, we observe that this aggressive retraining weakens the transferability of the learned representations

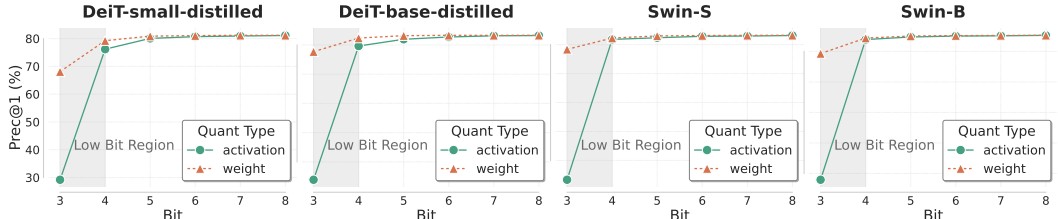

Figure 2: Impact of quantizing weights and activations separately.

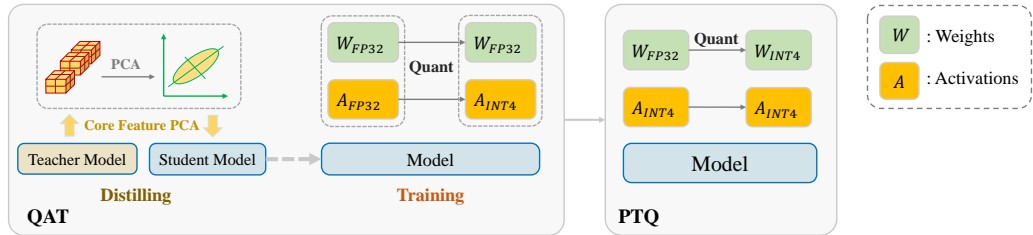

Figure 3: Overview of GPLQ: QAT stage first only for activations, then PTQ stage only for weights.

to downstream tasks (cf. Table 1 and Figure 1). Our second insight is to *make the quantized model stay in the same basin*. This is achieved in GPLQ by i) restricting QAT to optimize only activation quantizers; ii) use a low learning rate and only 1 epoch training in this QAT stage; iii) a feature mimicking loss that encourages the quantized model to retain the key feature structures of the original FP32 model.

## 3.2 The GPLQ Framework

### 3.2.1 Stage 1: Activation-only Quantization-Aware Training, or Act-QAT

This stage only quantizes activations. The key is that all model weights are *learnable but kept at FP32 precision*, thereby decoupling activation quantizer learning from weight updates and effectively circumventing weight oscillation that occurs in QAT which quantizes the two simultaneously.

Activations employ the uniform symmetric quantization implemented with per-channel granularity. A comparison of per-tensor versus per-group quantization is detailed in the appendix. Activations are quantized to 4 bits ($b = 4$) in our experiments, but our method and code work for lower-bit quantization, too. The quantization scaling factor $s_a$ is learned using LSQ [7]. One *novel proposal in GPLQ* is that we initialize $s_a$ and calibrate it on a small subset of training data using a percentile-based min-max PTQ method. For internal per-token quantized operations (if applicable), the initial quantization range is set based on the 1st and 99th percentiles of observed activation values to mitigate outlier effects; for per-channel quantization, min-max values are used directly. This initialization process is fast (typically in seconds) and provides a better starting point for subsequent LSQ-like optimization. The quantized activation $\hat{x}$ for an input $x$ is calculated as follows:

$$\hat{x} = \text{clamp}(\text{round}(x/s_a), -2^{b-1}, 2^{b-1} - 1) \times s_a \,, \tag{1}$$

where $b = 4$ in our experiments. The gradient of $s_a$ is estimated using Straight-Through Estimator (STE).

To preserve the rich representational power of pre-trained models (i.e., keep the generalization power), we follow a lightweight feature mimicking loss inspired by TCS [40].We chose the penultimate layer based on two considerations: 1) **High-level semantics:** This layer's features contain the highest level of semantic information, crucial for generalization. 2) **Efficiency:** Constraining features from a single layer minimizes computational overhead, aligning with our "lightning-fast" principle. We first extract features $f_t$ from the penultimate layer of the original FP32 teacher model. Then, PCA is performed on a set of teacher features $F_t = \{f_{t_i}\}$ obtained from a subset of the training data. The resulting principal components $V$ define a low-dimensional subspace that captures the main information in the teacher's feature manifold, which is consistent with findings that transformer

features are inherently low-rank [37]. We select principal components that explain a majority of the variance in $F_t$ (e.g., approximately 60%), and for hardware friendliness, adjust the number of selected principal components to be a multiple of 32. Specifically, for Swin-T [24] with 768 dimensions, 256 are selected; for DeiT-T [32] with 192 dimensions, 64 are selected. The corresponding features $f_s$ of the student model (the model undergoing Act-QAT) are projected onto this PCA-defined subspace. The loss $L_{PCA}$ is defined as the Mean Squared Error (MSE) between the student's projected features and the teacher's projected features, i.e., matching projections in the PCA space:

$$L_{PCA} = \frac{1}{N} \sum_{i=1}^{N} ||(f_s^i - \mu_t)V_{sel} - (f_t^i - \mu_t)V_{sel}||_2^2, \tag{2}$$

where $\mu_t$ is the mean of the teacher features in $F_t$, $V_{sel}$ are the selected principal components, and $N$ is the number of samples (e.g., batch size). Here, $\mu_t$ is the mean of the teacher features and cannot be eliminated. In PCA projection, data must be centered by subtracting the mean before projection. Both student ($f_s^i$) and teacher ($f_t^i$) features are centered using the *teacher's* mean ($\mu_t$) to ensure they are compared in the same coordinate system. This loss encourages the post-quantization activations to compactly retain salient features of the FP32 teacher model, thereby enhancing generalization. In our experiments, the weight ratio of the feature-mimicking loss ($L_{PCA}$) to the original classification loss was set to 1:1, avoiding an additional hyperparameter. An ablation study on this weight is provided in the Appendix.

### 3.2.2 Stage 2: Post-Training Quantization of Weights, or Weight-PTQ

Now the model has FP32 weights and 4-bit quantized activations (W32A4). The second stage rapidly quantizes the weights to 4 bits using mature PTQ techniques to generate the final W4A4 model. We leverage existing efficient PTQ methods in this stage. Specifically, after quantizing the weights using RepQ-ViT [20], we further apply the QwT (Quantization without Tears) [9, 31] method to compensate for the accuracy loss introduced by weight quantization. The entire calibration process is completed on a small, randomly selected subset of the ImageNet training data.

With activation quantizers *frozen*, we significantly simplify PTQ of weights. Since activations are already fixed and quantized to 4 bits (A4), error coupling that hurts weight PTQ algorithm is greatly reduced. The primary source of error is weight quantization itself (W32 $\to$ W4), rather than the compound error in simultaneous W32 $\to$ W4 and A32 $\to$ A4 conversions in traditional PTQ. This simplified objective makes the compensation technique (QwT) easier to perform. QwT corrects quantization error by introducing a lightweight linear compensation layer, whose parameters $W^*$ are determined by the following closed-form solution:

$$W^* = (Y - Y_Z)X_Z^T(X_Z X_Z^T + \lambda I)^{-1}, \tag{3}$$

which includes a regularization term $\lambda I$ for stability. In our W32A4 setting, $X_Z$ represents the input activations that have already been quantized to 4 bits, $Y$ is the output of the layer with FP32 weights and A4 inputs ($Y_{W32A4}$), while $Y_Z$ is the output with 4-bit weights and A4 inputs ($Y_{W4A4}$).

### 3.3 Advantages of Our GPLQ

GPLQ is designed with practical ease of use in mind, and we provide an implementation that encapsulates the two-stage process into simple and easy-to-use code.

The principles and effectiveness of GPLQ are not limited to image classification. We extended the code to object detection on MS-COCO [22]. For COCO tasks, Act-QAT (Stage 1) is also performed for 1 epoch, but due to larger input resolutions and higher model complexity the batch size per GPU is adjusted to 1. The subsequent weight PTQ (Stage 2) follows a similar procedure to that for image classification. This demonstrates the good generality of our framework across diverse vision tasks.

Compared to traditional QAT and PTQ methods, GPLQ offers an attractive alternative. Requiring only 1 epoch of activation QAT, its training duration and required computational resources are far less than those of a typical full QAT process (e.g., *hundreds of times faster*). Furthermore, pre-quantized activations create a more tractable optimization problem for subsequent weight PTQ, thereby enhancing the effectiveness of methods like QwT and the weight quantization in RepQ-ViT. Finally, our design avoids jumping out of the FP32 model's local minima, thus is useful for preserving the generalization ability of the pre-trained FP32 model.

Table 1: Comparison (Top-1 accuracy in percentage) between GPLQ and SOTA QAT methods on ImageNet-1k and 5 downstream FGVC tasks. 'Avg Task' is the average accuracy on 5 FGVC tasks.

| Network | Method | Mode | ImageNet | Aircraft | Food101 | Flowers102 | Pets | Cars | Avg Task |
|---|---|---|---|---|---|---|---|---|---|
| Swin-T | FP32 | W32A32 | 81.2 | 39.72 | 73.85 | 91.10 | 93.21 | 56.06 | 70.79 |
| | OFQ [23] | W4A4 | 81.9 | 26.58 | 64.79 | 84.40 | 91.74 | 36.13 | 60.73 |
| | Q-Var [15] | W4A4 | **82.4** | 37.02 | 70.98 | 87.15 | 92.86 | 52.57 | 68.12 |
| | RepQ-ViT [20] | W4A4 | 73.0 | 35.46 | 60.59 | 86.83 | 88.74 | 42.33 | 62.79 |
| | **GPLQ** | W4A4 | 79.5 | **40.80** | **73.15** | **93.10** | **93.32** | **53.46** | **70.77** |
| Swin-S | FP32 | W32A32 | 83.2 | 38.13 | 72.63 | 91.32 | 92.80 | 55.48 | 70.07 |
| | RepQ-ViT [20] | W4A4 | 71.9 | 32.16 | 69.44 | 90.40 | 93.08 | 49.27 | 66.87 |
| | **GPLQ** | W4A4 | **82.0** | **35.73** | **74.56** | **91.75** | **93.49** | **52.42** | **69.60** |
| Swin-B | FP32 | W32A32 | 85.3 | 49.71 | 85.12 | 99.48 | 94.49 | 65.86 | 78.93 |
| | RepQ-ViT [20] | W4A4 | 69.0 | 44.85 | 63.44 | 95.74 | 89.13 | 59.25 | 70.48 |
| | **GPLQ** | W4A4 | **84.0** | **49.65** | **83.81** | **99.56** | **94.03** | **63.91** | **78.25** |
| Swin-L | FP32 | W32A32 | 86.3 | 51.52 | 87.10 | 99.63 | 94.93 | 71.25 | 80.89 |
| | RepQ-ViT [20] | W4A4 | 83.2 | 52.66 | 84.84 | 99.46 | 94.36 | 67.31 | 79.73 |
| | **GPLQ** | W4A4 | **85.3** | **52.97** | **87.66** | **99.58** | **94.71** | **69.17** | **80.82** |
| DeiT-S | FP32 | W32A32 | 81.2 | 34.41 | 64.92 | 87.36 | 91.85 | 50.50 | 65.81 |
| | OFQ [23] | W4A4 | **81.1** | 29.07 | 65.88 | 83.61 | 92.10 | 42.31 | 62.59 |
| | RepQ-ViT [20] | W4A4 | 72.7 | 26.55 | 57.68 | 85.28 | 89.67 | 40.59 | 59.95 |
| | QwT [9] | W4A4 | 74.8 | 35.61 | 61.23 | 87.75 | 88.93 | 48.09 | 64.32 |
| | **GPLQ** | W4A4 | 78.8 | **39.81** | **66.55** | **89.77** | **92.45** | **50.55** | **67.83** |
| DeiT-B | FP32 | W32A32 | 83.3 | 45.06 | 72.96 | 91.84 | 93.35 | 63.93 | 73.43 |
| | RepQ-ViT [20] | W4A4 | 76.3 | 48.90 | 69.47 | 93.35 | 92.56 | 62.39 | 73.33 |
| | QwT [9] | W4A4 | 78.5 | 49.23 | 73.37 | 93.85 | 92.59 | **65.96** | 75.00 |
| | **GPLQ** | W4A4 | **82.0** | **50.89** | **73.37** | **94.28** | **93.27** | 64.98 | **75.36** |

# 4 Experiments

We conducted a comprehensive evaluation of GPLQ on multiple benchmark datasets and vision tasks. For image classification, we use ImageNet-1k [4] for pre-training and primary performance evaluation. For object detection and instance segmentation tasks, we employ the COCO 2017 [22] dataset, by training models on the 'train2017' set and reporting performance on the 'val2017' set.

To evaluate the model's generalization ability, we also selected five commonly used Fine-Grained Visual Classification (FGVC) datasets, including Aircraft [27], Food101 [2], Flowers102 [28], Pets [30], and Cars [17]. To fairly compare the feature extraction capabilities and downstream generalization performance, we train models using linear probing. Furthermore, for a fair comparison, the hyperparameters used for all methods were kept consistent following DTL [10]: 100 epochs, learning rate 0.001, batch size of 64, and drop path rate 0.1.

Stage 1 (Act-QAT) trained for 1 epoch on ImageNet-1k (classification) or COCO (detection) using AdamW [26] with a fixed learning rate of $5 \times 10^{-6}$ and no decay. Activations use per-channel symmetric 4-bit quantization. The subspace dimension used for PCA feature mimicking is dynamically selected based on the model's feature dimension, with the selection primarily based on accumulated variance when the accumulated variance is around 60%. Specifically, Swin-T uses a 256-dimensional PCA subspace, and DeiT-T (with 192 dimensions) uses a 64-dimensional subspace. Training was conducted on 8 GPUs, with a batch size of 16 per GPU. This configuration allows quantizing of the entire Swin Transformer series even on consumer-grade GPUs.

Stage 2 (Weight-PTQ) employed percentile-based per-channel symmetric 4-bit quantization for weights, combined with QwT [9] for compensation. The calibration set consists of 512 randomly selected images from the ImageNet training set.

## 4.1 Image Classification Performance

We evaluated GPLQ on ImageNet-1k and five downstream fine-grained classification tasks. The results are shown in Table 1, from which we can observe that:

Table 2: Object detection and instance segmentation results ($AP^{box}$ / $AP^{mask}$).

| Method | Bits (W/A) | Swin-T (1x) | Swin-T (3x) | Swin-S (3x) |
|---|---|---|---|---|
| Full-Precision | 32/32 | 0.426 / 0.393 | 0.460 / 0.416 | 0.485 / 0.433 |
| PTQ4ViT [39]) | 4/4 | — / — | 0.069 / 0.070 | 0.267 / 0.266 |
| APQ-ViT [7] | 4/4 | — / — | 0.237 / 0.226 | 0.447 / 0.401 |
| RepQ-ViT [20] | 4/4 | 0.135 / 0.137 | 0.361 / 0.360 | 0.426 / 0.400 |
| GPLQ (Act-QAT only) | 32/4 | 0.397 / 0.381 | 0.430 / 0.402 | 0.457 / 0.421 |
| **GPLQ** | 4/4 | **0.379 / 0.368** | **0.401 / 0.389** | **0.434 / 0.413** |

1. **On ImageNet Itself.** GPLQ significantly outperforms PTQ methods (RepQ-ViT and QwT). On Swin-T, GPLQ is 6.5% higher than RepQ-ViT and only 1.7% lower than FP32. For DeiT-S, GPLQ also far surpasses RepQ-ViT and QwT. Compared to QAT methods (OFQ, Q-Var), GPLQ is slightly inferior to these QAT models. But, these QAT methods not only require *unacceptable training time* but *may also lead to overfitting* (which we discuss next).

2. **Downstream Task Generalization:** In terms of average downstream accuracy, GPLQ almost always surpasses other quantization methods. And it consistently achieves better generalization than FP32. Swin-T GPLQ's average downstream accuracy is 70.77%, nearly identical to FP32's 70.79%, and far exceeding Q-Var (68.12%) and OFQ (60.73%), We want to emphasize that *QAT models, although exhibiting highest accuracy on ImageNet, lag behind PTQ methods in terms of generalization* (downstream accuracy). On the other hand, our *GPLQ has clearly better generalization than PTQ methods*, despite being a QAT method.

3. Due to the limited open-source availability of QAT methods, the number of compared QAT methods is small. And, QAT methods compared in this paper are only for small models, because they are out-of-memory even for medium size models. Our GPLQ *scales to large models and will be open-source*.

4. Q-Var performs better than OFQ in generalization, because it uses an external EfficientNet-L2 (88.2% accuracy on ImageNet) pre-trained on JFT-300M as a teacher. Finding an equally powerful teacher in other tasks is impossible or difficult and limits its application scenarios.

In short, our GPLQ quantizes super-fast, has both high accuracy and excellent generalization.

## 4.2 Object Detection Performance

We also evaluated GPLQ for object detection using the Mask R-CNN [12]framework on COCO 2017. Here GPLQ did not employ PCA feature mimicking. Results are shown in Table 2.

For a fair comparison, our weight quantization uses the same weight quantization method as in RepQ-ViT to complete the second stage. Although there is some degradation compared to W32A4 (our Stage 1 model), GPLQ still significantly outperforms other W4A4 methods such as RepQ-ViT, PTQ4ViT, and APQ-ViT. On Swin-T (3x), GPLQ achieves 0.401 $AP^{box}$, while RepQ-ViT only achieves 0.361. Even without using PCA feature mimicking, the core two-stage idea of GPLQ still demonstrates strong competitiveness in object detection tasks, which shows its generality.

## 4.3 Ablation Studies

We first investigate the impact of different activation quantization granularities (per-channel vs. per-layer) during the Act-QAT stage. As shown in Table 3, using per-channel activation quantization consistently outperforms per-layer quantization in both ImageNet and average downstream accuracy. Notably, even with per-layer quantization, our GPLQ can still achieve generalization performance close to the original floating-point model in most cases. For example, Swin-T (Layer-wise) achieved an 'Avg Task' of 68.7%, only 2.1% lower than FP32's 70.8%. Per-channel quantization, meanwhile, maintained generalization performance almost identical to that of FP32 across all models.

**Contribution of different components in the second stage of GPLQ.** We first use the first stage Act-QAT to obtain a W32A4 model. Then, we apply different PTQ strategies to the weights of this model. Results are shown in Table 4. It can be observed that even when only using the basic

Table 3: Ablation on Act-QAT granularities.

| Model (Swin) | Prec. (W/A) | ImageNet Top-1 Acc. (%) | Avg Task (%) |
|---|---|---|---|
| Swin-T FP32 | W32A32 | 81.2 | 70.8 |
| Channel wise | W4A4 | 79.5 (-1.7) | 70.8 (=) |
| Layer wise | W4A4 | 77.9 (-3.3) | 68.7 (-2.1) |
| Swin-S FP32 | W32A32 | 83.2 | 70.1 |
| Channel wise | W4A4 | 82.0 (-1.2) | 69.6 (-0.5) |
| Layer wise | W4A4 | 81.3 (-1.9) | 67.8 (-2.3) |
| Swin-B FP32 | W32A32 | 85.3 | 78.9 |
| Channel wise | W4A4 | 84.0 (-1.3) | 78.2 (-0.7) |
| Layer wise | W4A4 | 82.7 (-2.5) | 72.4 (-6.5) |
| Swin-L FP32 | W32A32 | 86.3 | 80.9 |
| Channel wise | W4A4 | 85.3 (-1.0) | 80.8 (-0.1) |
| Layer wise | W4A4 | 84.9 (-1.4) | 79.0 (-1.9) |

Table 4: Ablation on weight-PTQ components.

| Model (Swin) | Operation | Prec. (W/A) | ImageNet Top-1 Acc. (%) | Avg Task (%) |
|---|---|---|---|---|
| Swin-T | FP32 | W32A32 | 81.2 | 70.8 |
| | GPLQ Act-QAT | W32A4 | 80.2 (-1.0) | 71.1 (+0.3) |
| | + PTQ weight | W4A4 | 79.0 (-2.2) | 69.0 (-1.8) |
| | + QwT [9] | W4A4 | 79.5 (-1.4) | 70.8 (=) |
| Swin-S | FP32 | W32A32 | 83.2 | 70.1 |
| | GPLQ Act-QAT | W32A4 | 82.4 (-0.8) | 70.5 (+0.4) |
| | + PTQ weight | W4A4 | 81.6 (-1.6) | 68.9 (-1.2) |
| | + QwT [9] | W4A4 | 82.0 (-1.2) | 69.6 (-0.5) |
| Swin-B | FP32 | W32A32 | 85.3 | 78.9 |
| | GPLQ Act-QAT | W32A4 | 84.4 (-0.9) | 78.8 (-0.1) |
| | + PTQ weight | W4A4 | 83.7 (-1.6) | 77.3 (-1.6) |
| | + QwT [9] | W4A4 | 84.0 (-1.3) | 78.2 (-0.7) |

Table 5: Optimizing basin retention.

| Medthod | Quantized model ImageNet(%) | Internal FP32 ImageNet(%) | Downstream Avg(%) |
|---|---|---|---|
| Raw FP32 | 81.2 | 81.2 | 70.79 |
| Q-Variation [15] | 82.4 | 68.9 | 68.12 |
| **GPLQ** | **79.5** | **81.1** | **70.77** |

Table 6: PCA Dimensionality.

| Method | PCA dim | ImageNet (%) | Avg Task Top-1 (%) |
|---|---|---|---|
| w/o PCA | – | 79.4 | 69.25 |
| w/ PCA | 64 | 79.5 | 70.12 |
| W/ PCA | 256 | 79.5 | 70.77 |
| W/ PCA | 512 | 79.4 | 70.36 |

percentile-based PTQ method to directly quantize weights to 4 bits (PTQ weight), the resulting W4A4 model is clearly worse than the W32A4 model, but is already within an acceptable range, indicating that our Act-QAT lays a good foundation for subsequent weight quantization. Furthermore, when QwT is applied for compensation after weight quantization ('+QwT'), the model's accuracy loss is significantly mitigated. The W4A4 model with QwT achieved ImageNet accuracy close to W32A4, and its average downstream accuracy was almost identical to the original FP32 model. This demonstrates the effectiveness of our two-stage method, where activations are first stabilized, then weights are quantized and supplemented with lightweight compensation.

**Importance of Preserving Optimization Basin for Generalization.** We designed an experiment to validate our hypothesis: if QAT methods significantly deviate from the original FP32 optimization basin, generalization ability may be impaired. We compared the FP32 model, a QAT method (using Quantization-Variation as an example), and our GPLQ. For QAT and GPLQ, we extracted their trained FP32 weights (i.e., remove quantization nodes and use the learned "latent" FP32 weights, denoted as 'Internal FP32') and evaluated their performance on ImageNet.

As shown in Table 5, although Quantization-Variation achieved a high quantized model accuracy on ImageNet (82.4%), the performance of its internal FP32 weights dropped significantly to 68.9%, far below FP32's 81.2%. This indicates that, the quantized weights have significantly deviated from the original optimization basin, potentially led to overfitting to the ImageNet train and val set, and ultimately impaired its generalization ability on downstream tasks (Avg Acc. 68.12%). Figure 4a compares the loss curves of GPLQ and traditional QAT. GPLQ exhibits a smooth convergence process without severe oscillations, indicating that it remains within the original optimization basin of the FP32 model. In contrast, QAT's loss soars to the level for a randomly initialized network (above 7.0) at the very beginning of training—it jumped out of the original local minimum and started to fit the specifics of ImageNet, thereby affecting final generalization adversely.

In contrast, GPLQ quantized model has an accuracy of 79.5%, and the accuracy of its extracted FP32 weights is 81.1%, which is very close to the original FP32 model. This fact strongly suggests that GPLQ successfully stays near its original optimization basin. This strategy not only achieves high accuracy on ImageNet, more critically, it maintains excellent downstream task generalization (Avg Acc. 70.77%), matching the FP32 model.

**Impact of PCA Dimensionality in Feature Distillation.** To further investigate the effectiveness of the PCA feature distillation module in GPLQ, we conducted ablation experiments specifically on the choice of PCA projection dimensionality on the Swin-T model. Except for whether PCA projection learning was used, all other settings followed GPLQ's default settings. We report its Top-1 accuracy

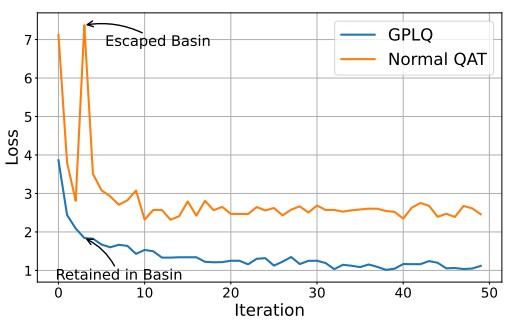 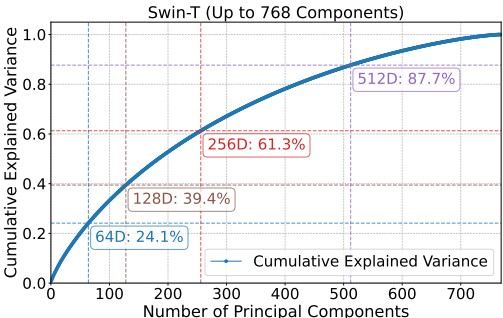

(a) Comparison of Training Loss Curves  (b) Swin-T PCA Cumulative Explained Variance

Figure 4: Training loss curves and Percentage of explained variance in GPLQ.

on the ImageNet validation set and the average accuracy from linear probing on 5 downstream FGVC datasets. Results are shown in Table 6.

We observed that without PCA feature mimicking ('Without PCA'), the model's average downstream task accuracy was 69.25%. When PCA feature mimicking was introduced, even with few dimensions (64), downstream task performance improved (70.12%). As the PCA dimensionality increased (256, at which point the cumulative explained variance was about 60%), the average downstream accuracy reached 70.77%. Further increasing the dimensionality to 512 resulted in a slight drop to 70.36%, but it is still better than not using PCA. These results indicate that feature mimicking can effectively guide the quantized model to learn key features from the FP32 model, thereby enhancing its generalization ability. Selecting a dimension that captures around 60% of the original feature variance represents a good trade-off.

Figure 4b shows Swin-T model's PCA cumulative explained variance with varying dimensions. At about 256 dimensions, the cumulative explained variance reaches approximately 61.3%. Even with 64 dimensions where only 24.1% variance explained, PCA feature mimicking already shows better generalization ability than without using PCA feature mimicking, as shown in Table 6.

**Quantitative Analysis of Stability and Decoupling.** To provide deeper quantitative evidence for our "same basin" and "activation-first" hypotheses, we conducted two further analyses. First, we tracked the weight deviation (Mean Squared Error) between the original FP32 weights and the model's weights during training. As shown in Table 7, GPLQ's Act-QAT stage keeps the weights extremely close to the original (0.049 MSE), as weights are in FP32 and only activations are quantized. In contrast, a "Direct W4A4" QAT (quantizing both W & A for 1 epoch) immediately diverges, with a deviation nearly 6 times larger (0.289 MSE). This confirms GPLQ's design prevents the model from "jumping out" of the original basin.

Table 7: Average per-layer weight deviation (MAE / MSE) from original FP32 weights during 1 epoch of training (Swin-T). GPLQ's decoupled strategy maintains weights within the original optimization basin, while direct W4A4 QAT diverges significantly.

| Method | Iter. 100 | Iter. 1k | Final (1-epoch) |
|---|---|---|---|
| GPLQ (Act-QAT) | 0.082 / 0.046 | 0.083 / 0.047 | 0.086 / 0.049 |
| Direct W4A4 QAT | 0.256 / 0.271 | 0.257 / 0.274 | 0.269 / 0.289 |

Second, we analyzed the benefit of our sequential strategy. GPLQ's "activation-first" stage stabilizes activations, creating a simpler problem for the subsequent weight PTQ. We measured the output error (MSE) introduced *only* by weight PTQ in two scenarios: 1) Standard PTQ (W32A32 → W4A4) and 2) GPLQ's Stage 2 (W32A4 → W4A4). Table 8 shows that because GPLQ has already stabilized the activations, the subsequent weight quantization error is dramatically reduced—by nearly **46%** on average for Swin-T. This demonstrates that our decoupled strategy is superior because it stabilizes activations first, making the final weight quantization step inherently easier and more accurate.

Table 8: PTQ Error (Average Per-Layer Output MAE / MSE) with (GPLQ) vs. without (Standard) pre-stabilized activations. Stabilizing activations first (GPLQ) cuts the subsequent weight quantization error almost in half.

| PTQ Scenario | Model | Layer1 block1 | Layer4 block2 | Avg. (All Layers) |
|---|---|---|---|---|
| Standard (on W32A32) | Swin-T | 0.359 / 0.239 | 0.426 / 0.530 | 0.184 / 0.136 |
| **GPLQ (on W32A4)** | Swin-T | **0.150 / 0.045** | **0.224 / 0.206** | **0.141 / 0.074** |
| Standard (on W32A32) | Swin-S | 0.188 / 0.058 | 0.302 / 0.368 | 0.164 / 0.140 |
| **GPLQ (on W32A4)** | Swin-S | **0.123 / 0.034** | **0.156 / 0.106** | **0.129 / 0.076** |

## 5 Conclusion

We presented GPLQ, a novel quantization framework for Vision Transformers that significantly improves efficiency and generalization over existing PTQ and QAT methods. Our approach is grounded in the empirical finding that activation quantization is critical and that preserving the original model's optimization basin is key to maintaining generalization. GPLQ's "activation-first, weights-later" strategy, featuring single-epoch activation quantization with PCA-based feature mimicking, followed by PTQ for weights, achieves 4-bit performance competitive with, and sometimes superior to, FP32 models in terms of generalization. This methodology not only drastically cuts training overhead, making advanced quantization more accessible, but also consistently outperforms prior art. GPLQ thus offers a practical and robust path for deploying low-bit ViTs in resource-constrained scenarios, facilitated by our forthcoming open-source toolkit.

## 6 Limitations and Future Work

Limitations include:

- Broader QAT comparison: Limited open-source availability of advanced QAT methods restricted a comprehensive comparative analysis.
- Dependence on PTQ techniques: GPLQ's second stage performance is tied to the capabilities and limitations of existing PTQ methods.
- Low bit-width exploration: Our work primarily focused on 4-bit quantization, mainly because 4-bit is where the hardware support ends at. Deeper investigation into even lower bit-widths is desired, too.

Future directions include:

- Extending PCA-based feature mimicking to further enhance model generalization across a broader range of vision tasks, such as object detection and semantic segmentation.
- Conducting more comprehensive and rigorous QAT benchmarks against a wider array of contemporary methods as they become publicly accessible and resources permitting.
- Adapting and evaluating the GPLQ framework for its applicability and effectiveness on diverse neural network architectures, including CNNs [13] and emerging LLMs [1, 33, 38].
- Integrating next-generation PTQ advancements to continually improve GPLQ's performance.

## Acknowledgments and Disclosure of Funding

This work was partly supported by the National Natural Science Foundation of China under Grant 62276123.

JW identified the problems and conjectures in QAT, and guided GL in conducting the experiments. During the experimental process, GL and JW jointly designed the GPLQ method. JW and GL wrote this paper. XL helped GL complete parts of the experiments and writing.

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

# A   Appendix

This appendix provides supplementary experimental results and analyses that further support the findings presented in the main paper.

## A.1   Quantization under Constrained Computational Resources

To assess the robustness and efficiency of our GPLQ method under more restrictive computational environments, we investigated the impact of varying the number of available GPUs during the activation quantization stage (Act-QAT). For these experiments, only activations were quantized for 1 epoch. The batch size per GPU was maintained at 16. Consequently, a reduction in the number of GPUs corresponds to a proportional decrease in the effective batch size and the learning rate was adjusted accordingly. Other training parameters remained consistent with the settings described in the main paper.

Table 9 details the performance of Swin-T on ImageNet and the associated training times.

Table 9: Impact of varying GPU counts on Swin-T (W32A4 Act-QAT only) ImageNet accuracy and training time. Batch size per GPU is 16.

| Number of GPUs | Equivalent Batch Size | Learning Rate | ImageNet Acc (%) | Time (min) |
|---|---|---|---|---|
| 8 | 128 (16×8) | 5e-6 | 80.2 | 35 |
| 4 | 64 (16×4) | 2.5e-6 | 80.1 | 71 |
| 2 | 32 (16×2) | 1e-6 | 80.1 | 139 |
| 1 | 16 (16×1) | 1e-6 | 80.1 | 275 |

The results indicate that GPLQ maintains high accuracy on ImageNet even as the number of GPUs and, consequently, the effective batch size and learning rate, are significantly reduced. The performance remains remarkably stable (80.1-80.2% top-1 accuracy) across all tested configurations. This resilience suggests that our method can effectively train quantized models even with minimal training resources, a capability not typically demonstrated by traditional QAT methods. Traditional QAT approaches often rely on large batch sizes and learning rates for stable convergence, and their performance is expected to degrade under such resource-constrained conditions. Our findings underscore the practical advantage of GPLQ in scenarios with limited hardware availability.

## A.2   Impact of Training Data Volume on Model Performance

We further analyzed the influence of the training data volume on the Swin-T model's performance during the 1-epoch activation quantization stage. All training settings were kept consistent with the GPLQ defaults, except for the number of training images used, which was varied by controlling the number of training iterations.

Table 10: Impact of training data volume (number of images used in 1 epoch of Act-QAT) on Swin-T ImageNet Top-1 accuracy (%) for W32A4 (only activations QAT) and W4A4 (activations QAT, weights PTQ) settings.

| Training Iterations | Images Used | W32A4 Acc (%) | W4A4 Acc (%) |
|---|---|---|---|
| 1 | 128 | 70.9 | 68.5 |
| 10 | 1,280 | 74.1 | 72.5 |
| 100 | 12,800 | 77.9 | 77.0 |
| 1000 | 128,000 | 79.6 | 79.0 |
| 10009 | Full Dataset (approx. 1.28M) | 80.2 | 79.5 |

As expected, increasing the volume of training data generally leads to improved model accuracy. Even with a relatively small number of images (e.g., 128,000, corresponding to about 10% of the full ImageNet training set for 1 epoch), the model achieves a respectable 79.0% accuracy. Training on the full dataset for 1 epoch yields 80.2% accuracy for the W32A4 model (output of Act-QAT stage), which forms a strong basis for the subsequent weight PTQ stage.

## A.3 Impact of Direct W4A4 Training versus Sequential Quantization

To further highlight the benefits of our proposed sequential "activation-first, weights-later" (W32A4 → W4A4) strategy, we compare it against a more direct approach where both weights and activations are quantized simultaneously from the start of the 1-epoch QAT process (direct W4A4). All other training hyperparameters for this direct W4A4 baseline were kept identical to those used in the Act-QAT stage of our GPLQ method.

Table 11 presents the ImageNet top-1 accuracy for various Swin Transformer models.

Table 11: Comparison of ImageNet top-1 accuracy (%) for Swin Transformers using direct W4A4 QAT versus GPLQ's sequential (W32A4 → W4A4) approach. Optimal results are in **bold**.

| Training Method | Swin-T | Swin-S | Swin-B | Swin-L |
|---|---|---|---|---|
| Direct W4A4 (1 epoch QAT) | 78.5 | 81.2 | 83.5 | 84.9 |
| GPLQ (W32A4 → W4A4) | **79.5** | **82.0** | **84.0** | **85.3** |

The results clearly demonstrate that our proposed sequential quantization strategy (GPLQ) consistently outperforms the direct W4A4 QAT approach across all Swin Transformer variants. As discussed in the main paper, quantizing activations first (while keeping weights in FP32) and then applying PTQ to the weights offers several advantages. Beyond the slight improvements in training speed and reduced memory footprint during the Act-QAT stage (as pseudo-quantization of weights is not performed), this sequential approach helps to avoid the weight oscillations often encountered in traditional QAT. This leads to a smoother optimization process for the model weights, ultimately resulting in improved final model performance, as evidenced by the higher accuracies in Table 11.

## A.4 Efficiency and Resource Overhead Comparison

Table 12 quantifies the "Practical" and "Lightning" advantages of GPLQ, as highlighted in Figure 1. Compared to traditional QAT methods like Q-Var and OFQ, GPLQ is over **500 times faster** (35 minutes vs. ~12 days). Furthermore, its peak memory footprint (6.6GB) is dramatically lower—even **less than standard FP32 training** (7.2GB)—and far below the prohibitive requirements of other QAT methods (25.9-37.5GB). This demonstrates GPLQ's practicality for real-world deployment on consumer-grade hardware.

Table 12: Detailed comparison of performance and resource overhead on ImageNet using a Swin-T model.

| Method | ImageNet Acc (%) | Avg. Downstream Acc (%) | Training Time (min) | Peak Memory (GB) |
|---|---|---|---|---|
| Full-precision (FP32) | 81.2 | 70.79 | 3078 (train from scratch) | 7.2 |
| RepQViT (PTQ) | 73.0 | 62.79 | 3 (calibration only) | 2.3 |
| Q-Var (QAT) | 82.4 | 68.12 | 17400 (~12 days) | 25.9 |
| OFQ (QAT) | 81.9 | 60.73 | 18360 (~12.7 days) | 37.5 |
| **GPLQ (Ours)** | **79.5** | **70.77** | **35 (1 epoch)** | **6.6** |

## A.5 Generality to CNN Architectures

While GPLQ was motivated by ViTs, its core principles—the importance of activations and preserving the optimization basin—are architecture-agnostic. To demonstrate its generality, we applied GPLQ to the classic ResNet [13] family. As shown in Table 13, GPLQ (W4A4) significantly outperforms the strong PTQ baseline (RepQViT) on all ResNet models and consistently improves downstream generalization with only 1 epoch of training.

## A.6 Impact of Act-QAT Training Duration

A key claim of GPLQ is its "Lightning" speed, achieved by training Act-QAT for only 1 epoch. We conducted an ablation study (Table 14) to validate this choice. Extending the training from 1 epoch to 10 epochs yields negligible performance gains (≤0.3%) on both ImageNet and downstream tasks,

Table 13: Generalization performance of GPLQ (W4A4) on CNN architectures (ResNets).

| Model | Method | ImageNet Acc (%) | Avg. Downstream Acc (%) | Training Epochs |
|---|---|---|---|---|
| ResNet-18 | FP32 | 69.10 | 58.78 | - |
| | RepQViT | 56.97 | 56.70 | - |
| | **GPLQ** | **66.16** | **59.44** | **1** |
| ResNet-34 | FP32 | 74.90 | 59.57 | - |
| | RepQViT | 64.76 | 54.13 | - |
| | **GPLQ** | **70.36** | **60.76** | **1** |
| ResNet-50 | FP32 | 78.70 | 66.17 | - |
| | RepQViT | 63.68 | 52.09 | - |
| | **GPLQ** | **74.13** | **67.68** | **1** |
| ResNet-101 | FP32 | 76.20 | 63.60 | - |
| | RepQViT | 56.97 | 60.26 | - |
| | **GPLQ** | **73.83** | **64.21** | **1** |

while increasing the training cost 10-fold. This supports 1 epoch as the optimal "sweet spot" for GPLQ's efficiency-performance trade-off.

Table 14: Impact of Act-QAT training duration on Swin-T performance.

| Act-QAT Epochs | ImageNet Top-1 Acc (%) | Avg. Downstream Acc (%) |
|---|---|---|
| **1 (Ours)** | 79.5 | 70.8 |
| 2 | 79.6 | 70.9 |
| 5 | 79.6 | 71.1 |
| 10 | 79.5 | 71.0 |

## A.7 Performance on Lower Bit-Widths (W3A3)

To test GPLQ's generality to more aggressive quantization, we evaluated it in a W3A3 setting (Table 15). GPLQ significantly outperforms the SOTA PTQ method (RepQ-ViT), e.g., by +21.5% on Swin-T. This demonstrates that GPLQ's "activation-first" framework is a powerful and general strategy, especially for challenging low-bit scenarios.

Table 15: Performance of GPLQ in a lower-bit (W3A3) setting.

| Model | Method | ImageNet Acc (%) | Avg. Downstream Acc (%) |
|---|---|---|---|
| Swin-T | FP32 | 81.2 | 70.79 |
| | RepQViT (W3A3) | 52.7 | 39.26 |
| | **GPLQ (W3A3)** | **74.2** | **65.76** |
| Swin-S | FP32 | 83.2 | 70.07 |
| | RepQViT (W3A3) | 65.9 | 53.35 |
| | **GPLQ (W3A3)** | **78.3** | **64.64** |
| Swin-B | FP32 | 85.3 | 78.93 |
| | RepQViT (W3A3) | 65.8 | 29.82 |
| | **GPLQ (W3A3)** | **80.3** | **74.15** |
| Swin-L | FP32 | 86.3 | 80.89 |
| | RepQViT (W3A3) | 71.3 | 66.36 |
| | **GPLQ (W3A3)** | **80.0** | **73.36** |

## A.8 Ablation on PCA Feature-Mimicking Loss Weight

As mentioned in Section 3.2.1, our default weight $\alpha$ for the $L_{PCA}$ loss is 1.0 (i.e., a 1:1 ratio with the main task loss). We conducted an ablation (Table 16) to validate this choice. The results show that $\alpha = 1.0$ provides the best trade-off, particularly for downstream generalization.

Table 16: Ablation analysis of the $L_{PCA}$ feature-mimicking loss weight ($\alpha$) on Swin-T.

| Loss Setting | Loss Weight ($\alpha$) | ImageNet Top-1 Acc (%) | Avg. Downstream Acc (%) |
|---|---|---|---|
| w/o PCA Loss | - | 79.4 | 69.2 |
| Penultimate | 0.1 | 79.5 | 70.3 |
| **Penultimate** | **1.0 (Default)** | **79.5** | **70.8** |
| Penultimate | 10 | 79.3 | 70.1 |

## A.9 Component Interchangeability

GPLQ is proposed as a flexible framework. To demonstrate this, we replaced the percentile-based initialization (from RepQ-ViT) in our PTQ stage with other common methods. As shown in Table 17, the framework is robust to this change, with final performance remaining high (79.2-79.5%). This confirms GPLQ's success stems from its "activation-first" paradigm, not a specific combination of components.

Table 17: Verifying the interchangeability of different PTQ initialization components within the GPLQ framework (Swin-T).

| Stage 2 (Weight-PTQ) Initialization | ImageNet Top-1 Acc (%) |
|---|---|
| **Percentile (as in RepQ-ViT)** | **79.5** |
| MSE-based | 79.5 |
| MinMax | 79.2 |

# NeurIPS Paper Checklist

1. **Claims**

   Question: Do the main claims made in the abstract and introduction accurately reflect the paper's contributions and scope?

   Answer: [Yes]

   Justification: In this paper, we claim in the abstract and introduction that GPLQ is a general, practical, and lightning-fast QAT method for ViTs, which outperforms existing PTQ methods and exhibits better generalization than traditional QAT approaches. Our main contributions, as stated, are: (1) new empirical insights regarding the paramount importance of activation quantization and the necessity of preserving the model's original optimization "basin"; (2) the GPLQ framework itself, which employs a novel "activation-first, weights-later" sequential quantization strategy; and (3) our commitment to release an easy-to-use open-source toolkit. These claims are substantiated by the experimental results presented in Section 4 and discussed throughout the paper. For example, Table 1 shows GPLQ's competitive accuracy and superior generalization, and Figure 1 highlights its efficiency.

   Guidelines:

   - The answer NA means that the abstract and introduction do not include the claims made in the paper.
   - The abstract and/or introduction should clearly state the claims made, including the contributions made in the paper and important assumptions and limitations. A No or NA answer to this question will not be perceived well by the reviewers.
   - The claims made should match theoretical and experimental results, and reflect how much the results can be expected to generalize to other settings.
   - It is fine to include aspirational goals as motivation as long as it is clear that these goals are not attained by the paper.

2. **Limitations**

   Question: Does the paper discuss the limitations of the work performed by the authors?

   Answer: [Yes]

   Justification: In Section 6, "Limitations and Future Work," we explicitly discuss the limitations of our work. These include the challenge of conducting a broader QAT comparison due to the limited open-source availability of advanced QAT methods, the dependence of GPLQ's second stage on existing PTQ techniques (inheriting their limitations), and the primary focus on 4-bit quantization, with a need for deeper exploration into lower bit-widths.

   Guidelines:

   - The answer NA means that the paper has no limitation while the answer No means that the paper has limitations, but those are not discussed in the paper.
   - The authors are encouraged to create a separate "Limitations" section in their paper.
   - The paper should point out any strong assumptions and how robust the results are to violations of these assumptions (e.g., independence assumptions, noiseless settings, model well-specification, asymptotic approximations only holding locally). The authors should reflect on how these assumptions might be violated in practice and what the implications would be.
   - The authors should reflect on the scope of the claims made, e.g., if the approach was only tested on a few datasets or with a few runs. In general, empirical results often depend on implicit assumptions, which should be articulated.
   - The authors should reflect on the factors that influence the performance of the approach. For example, a facial recognition algorithm may perform poorly when image resolution is low or images are taken in low lighting. Or a speech-to-text system might not be used reliably to provide closed captions for online lectures because it fails to handle technical jargon.
   - The authors should discuss the computational efficiency of the proposed algorithms and how they scale with dataset size.

- If applicable, the authors should discuss possible limitations of their approach to address problems of privacy and fairness.
- While the authors might fear that complete honesty about limitations might be used by reviewers as grounds for rejection, a worse outcome might be that reviewers discover limitations that aren't acknowledged in the paper. The authors should use their best judgment and recognize that individual actions in favor of transparency play an important role in developing norms that preserve the integrity of the community. Reviewers will be specifically instructed to not penalize honesty concerning limitations.

3. **Theory assumptions and proofs**

    Question: For each theoretical result, does the paper provide the full set of assumptions and a complete (and correct) proof?

    Answer: [NA]

    Justification: This paper is primarily empirical in nature. We present novel empirical insights (Section 3.1) and introduce the GPLQ framework (Section 3.2), validating our approach through extensive experiments (Section 4). While our method incorporates established mathematical formulations for quantization (Equation 1), PCA-based feature mimicking loss (Equation 2), and the QwT compensation mechanism (Equation 3), we do not propose new theoretical results that would necessitate formal proofs within this work.

    Guidelines:

    - The answer NA means that the paper does not include theoretical results.
    - All the theorems, formulas, and proofs in the paper should be numbered and cross-referenced.
    - All assumptions should be clearly stated or referenced in the statement of any theorems.
    - The proofs can either appear in the main paper or the supplemental material, but if they appear in the supplemental material, the authors are encouraged to provide a short proof sketch to provide intuition.
    - Inversely, any informal proof provided in the core of the paper should be complemented by formal proofs provided in appendix or supplemental material.
    - Theorems and Lemmas that the proof relies upon should be properly referenced.

4. **Experimental result reproducibility**

    Question: Does the paper fully disclose all the information needed to reproduce the main experimental results of the paper to the extent that it affects the main claims and/or conclusions of the paper (regardless of whether the code and data are provided or not)?

    Answer: [Yes]

    Justification: In Section 4 ("Experiments"), we provide comprehensive details for reproducing our main experimental results. This includes specifications of the datasets used (ImageNet-1k, COCO 2017, and five FGVC datasets), the evaluation metrics (Top-1 accuracy, $AP^{box}$, average downstream accuracy), and the detailed settings for both Stage 1 (Act-QAT) and Stage 2 (Weight-PTQ) of GPLQ. Key parameters such as the optimizer (AdamW), learning rates (fixed $5 \times 10^{-6}$ for Act-QAT), batch sizes (16 per GPU for Act-QAT), quantization specifics (4-bit symmetric, per-channel), PCA subspace dimensions (e.g., 256 for Swin-T, 64 for DeiT-T), and the calibration set size for PTQ (512 images) are explicitly stated. Furthermore, as noted in Contribution 3 (Section 1) and Section 3.3, we have also stated that "We will open-source GPLQ upon paper acceptance" and "GPLQ provides an easy-to-use quantization tool".

    Guidelines:

    - The answer NA means that the paper does not include experiments.
    - If the paper includes experiments, a No answer to this question will not be perceived well by the reviewers: Making the paper reproducible is important, regardless of whether the code and data are provided or not.
    - If the contribution is a dataset and/or model, the authors should describe the steps taken to make their results reproducible or verifiable.

- Depending on the contribution, reproducibility can be accomplished in various ways. For example, if the contribution is a novel architecture, describing the architecture fully might suffice, or if the contribution is a specific model and empirical evaluation, it may be necessary to either make it possible for others to replicate the model with the same dataset, or provide access to the model. In general. releasing code and data is often one good way to accomplish this, but reproducibility can also be provided via detailed instructions for how to replicate the results, access to a hosted model (e.g., in the case of a large language model), releasing of a model checkpoint, or other means that are appropriate to the research performed.
- While NeurIPS does not require releasing code, the conference does require all submissions to provide some reasonable avenue for reproducibility, which may depend on the nature of the contribution. For example
  (a) If the contribution is primarily a new algorithm, the paper should make it clear how to reproduce that algorithm.
  (b) If the contribution is primarily a new model architecture, the paper should describe the architecture clearly and fully.
  (c) If the contribution is a new model (e.g., a large language model), then there should either be a way to access this model for reproducing the results or a way to reproduce the model (e.g., with an open-source dataset or instructions for how to construct the dataset).
  (d) We recognize that reproducibility may be tricky in some cases, in which case authors are welcome to describe the particular way they provide for reproducibility. In the case of closed-source models, it may be that access to the model is limited in some way (e.g., to registered users), but it should be possible for other researchers to have some path to reproducing or verifying the results.

5. **Open access to data and code**

   Question: Does the paper provide open access to the data and code, with sufficient instructions to faithfully reproduce the main experimental results, as described in supplemental material?

   Answer: [Yes]

   Justification: In this paper, we explicitly state our intention to provide open access to our GPLQ framework. Specifically, in Contribution 3 (Section 1), we mention, "We will open-source GPLQ upon paper acceptance." This is reiterated in Section 3.3: "GPLQ provides an easy-to-use quantization tool supporting classification, detection and other downstream tasks." The datasets utilized in our experiments (ImageNet, COCO, and the various FGVC datasets) are all publicly available and widely used benchmarks, with citations provided.

   Guidelines:
   - The answer NA means that paper does not include experiments requiring code.
   - Please see the NeurIPS code and data submission guidelines (`https://nips.cc/public/guides/CodeSubmissionPolicy`) for more details.
   - While we encourage the release of code and data, we understand that this might not be possible, so "No" is an acceptable answer. Papers cannot be rejected simply for not including code, unless this is central to the contribution (e.g., for a new open-source benchmark).
   - The instructions should contain the exact command and environment needed to run to reproduce the results. See the NeurIPS code and data submission guidelines (`https://nips.cc/public/guides/CodeSubmissionPolicy`) for more details.
   - The authors should provide instructions on data access and preparation, including how to access the raw data, preprocessed data, intermediate data, and generated data, etc.
   - The authors should provide scripts to reproduce all experimental results for the new proposed method and baselines. If only a subset of experiments are reproducible, they should state which ones are omitted from the script and why.
   - At submission time, to preserve anonymity, the authors should release anonymized versions (if applicable).
   - Providing as much information as possible in supplemental material (appended to the paper) is recommended, but including URLs to data and code is permitted.

6. **Experimental setting/details**

   Question: Does the paper specify all the training and test details (e.g., data splits, hyper-parameters, how they were chosen, type of optimizer, etc.) necessary to understand the results?

   Answer: [Yes]

   Justification: We have detailed our experimental settings in Section 4 ("Experiments"). This includes the specific datasets used (ImageNet-1k, COCO 2017 with 'train2017' for training and 'val2017' for validation, and five FGVC datasets). For Stage 1 (Act-QAT), we specify 1 epoch of training, the AdamW optimizer, a fixed learning rate of $5 \times 10^{-6}$ without decay, per-channel symmetric 4-bit quantization for activations, PCA subspace dimensions (e.g., 256 for Swin-T, 64 for DeiT-T, chosen based on 60

   Guidelines:
   - The answer NA means that the paper does not include experiments.
   - The experimental setting should be presented in the core of the paper to a level of detail that is necessary to appreciate the results and make sense of them.
   - The full details can be provided either with the code, in appendix, or as supplemental material.

7. **Experiment statistical significance**

   Question: Does the paper report error bars suitably and correctly defined or other appropriate information about the statistical significance of the experiments?

   Answer: [No]

   Justification: In this paper, our experimental results, as presented in Tables 1, 2, 3, 4, and 5, report performance metrics (such as Top-1 accuracy, $AP^{box}$, and average downstream task accuracy) as single numerical values for each method and model configuration. We do not include error bars, confidence intervals, or explicit statistical significance tests (e.g., p-values) for these reported results.

   Guidelines:
   - The answer NA means that the paper does not include experiments.
   - The authors should answer "Yes" if the results are accompanied by error bars, confidence intervals, or statistical significance tests, at least for the experiments that support the main claims of the paper.
   - The factors of variability that the error bars are capturing should be clearly stated (for example, train/test split, initialization, random drawing of some parameter, or overall run with given experimental conditions).
   - The method for calculating the error bars should be explained (closed form formula, call to a library function, bootstrap, etc.)
   - The assumptions made should be given (e.g., Normally distributed errors).
   - It should be clear whether the error bar is the standard deviation or the standard error of the mean.
   - It is OK to report 1-sigma error bars, but one should state it. The authors should preferably report a 2-sigma error bar than state that they have a 96% CI, if the hypothesis of Normality of errors is not verified.
   - For asymmetric distributions, the authors should be careful not to show in tables or figures symmetric error bars that would yield results that are out of range (e.g. negative error rates).
   - If error bars are reported in tables or plots, The authors should explain in the text how they were calculated and reference the corresponding figures or tables in the text.

8. **Experiments compute resources**

   Question: For each experiment, does the paper provide sufficient information on the computer resources (type of compute workers, memory, time of execution) needed to reproduce the experiments?

   Answer: [Yes]

Justification: We provide information regarding the compute resources used for our experiments. In Section 4, we state: "Training was conducted on 8 GPUs, with a batch size of 16 per GPU. This configuration allows quantizing of the entire Swin Transformer series even on consumer-grade GPUs." We also highlight the efficiency of GPLQ: Act-QAT requires "only 1 epoch" (mentioned in Introduction and Section 2), is "hundreds of times faster" than traditional QAT (Introduction and Section 3.3), and reduces QAT training time "from days to 1-2 hours" (Section 1). This gives a clear indication of the hardware (8 GPUs, consumer-grade accessible) and the significant reduction in execution time compared to standard QAT methods. The memory footprint is also noted as "far lower than existing QAT methods" and "even lower than FP32 training" (Introduction, Section 1).

Guidelines:

- The answer NA means that the paper does not include experiments.
- The paper should indicate the type of compute workers CPU or GPU, internal cluster, or cloud provider, including relevant memory and storage.
- The paper should provide the amount of compute required for each of the individual experimental runs as well as estimate the total compute.
- The paper should disclose whether the full research project required more compute than the experiments reported in the paper (e.g., preliminary or failed experiments that didn't make it into the paper).

9. **Code of ethics**

Question: Does the research conducted in the paper conform, in every respect, with the NeurIPS Code of Ethics https://neurips.cc/public/EthicsGuidelines?

Answer: [Yes]

Justification: Our research, as presented in this paper, focuses on developing and evaluating a model quantization technique (GPLQ) for Vision Transformers. This is a standard and constructive area of machine learning research aimed at improving model efficiency. We have reviewed the NeurIPS Code of Ethics, and to the best of our knowledge, our work, including the methodology, data usage (publicly available academic datasets), and reporting, fully conforms to these guidelines. There are no aspects of our research that involve plagiarism, dual-use concerns beyond typical model efficiency improvements, fairness or privacy issues related to data subjects, or research with human subjects that would require special ethical considerations or approvals.

Guidelines:

- The answer NA means that the authors have not reviewed the NeurIPS Code of Ethics.
- If the authors answer No, they should explain the special circumstances that require a deviation from the Code of Ethics.
- The authors should make sure to preserve anonymity (e.g., if there is a special consideration due to laws or regulations in their jurisdiction).

10. **Broader impacts**

Question: Does the paper discuss both potential positive societal impacts and negative societal impacts of the work performed?

Answer: [No]

Justification: This paper primarily focuses on the technical contributions of GPLQ, detailing its methodology, efficiency, accuracy, and generalization capabilities for Vision Transformer quantization. While positive societal impacts are implicit in our work—such as enabling the deployment of powerful ViT models on resource-constrained devices, potentially leading to reduced energy consumption and wider accessibility of AI technologies—we have not dedicated a specific section to explicitly discuss these or potential negative societal impacts. The "Limitations and Future Work" section (Section 6) is confined to technical limitations and avenues for future research within the scope of model quantization.

Guidelines:

- The answer NA means that there is no societal impact of the work performed.
- If the authors answer NA or No, they should explain why their work has no societal impact or why the paper does not address societal impact.

- Examples of negative societal impacts include potential malicious or unintended uses (e.g., disinformation, generating fake profiles, surveillance), fairness considerations (e.g., deployment of technologies that could make decisions that unfairly impact specific groups), privacy considerations, and security considerations.
- The conference expects that many papers will be foundational research and not tied to particular applications, let alone deployments. However, if there is a direct path to any negative applications, the authors should point it out. For example, it is legitimate to point out that an improvement in the quality of generative models could be used to generate deepfakes for disinformation. On the other hand, it is not needed to point out that a generic algorithm for optimizing neural networks could enable people to train models that generate Deepfakes faster.
- The authors should consider possible harms that could arise when the technology is being used as intended and functioning correctly, harms that could arise when the technology is being used as intended but gives incorrect results, and harms following from (intentional or unintentional) misuse of the technology.
- If there are negative societal impacts, the authors could also discuss possible mitigation strategies (e.g., gated release of models, providing defenses in addition to attacks, mechanisms for monitoring misuse, mechanisms to monitor how a system learns from feedback over time, improving the efficiency and accessibility of ML).

11. **Safeguards**

Question: Does the paper describe safeguards that have been put in place for responsible release of data or models that have a high risk for misuse (e.g., pretrained language models, image generators, or scraped datasets)?

Answer: [NA]

Justification: Our work introduces GPLQ, a quantization method, and we plan to release an associated open-source toolkit. This toolkit is designed for model compression and efficiency, rather than being a pre-trained model with inherent high-risk misuse potential (like a large generative model) or a dataset containing sensitive information. Therefore, the specific safeguards typically associated with the release of such high-risk assets are not directly applicable to the GPLQ method or its toolkit as described in this paper. The models we quantize are standard Vision Transformers, and the datasets used are publicly available academic benchmarks.

Guidelines:

- The answer NA means that the paper poses no such risks.
- Released models that have a high risk for misuse or dual-use should be released with necessary safeguards to allow for controlled use of the model, for example by requiring that users adhere to usage guidelines or restrictions to access the model or implementing safety filters.
- Datasets that have been scraped from the Internet could pose safety risks. The authors should describe how they avoided releasing unsafe images.
- We recognize that providing effective safeguards is challenging, and many papers do not require this, but we encourage authors to take this into account and make a best faith effort.

12. **Licenses for existing assets**

Question: Are the creators or original owners of assets (e.g., code, data, models), used in the paper, properly credited and are the license and terms of use explicitly mentioned and properly respected?

Answer: [No]

Justification: In this paper, we have made sure to properly credit the original creators of the assets we used by citing their respective publications. This includes the datasets (e.g., ImageNet [4], COCO [22], various FGVC datasets [27, 2, 28, 30, 17]) and influential methods or models that form the basis or comparison for our work (e.g., DeiT [32], Swin-T [24], RepQ-ViT [20], QwT [9]). However, we have not explicitly stated the specific licenses (e.g., CC-BY 4.0, MIT License) or detailed terms of use for these pre-existing

assets within the body of our paper. We have operated under the assumption that their use for academic research purposes is permitted by their standard terms of availability.

Guidelines:

- The answer NA means that the paper does not use existing assets.
- The authors should cite the original paper that produced the code package or dataset.
- The authors should state which version of the asset is used and, if possible, include a URL.
- The name of the license (e.g., CC-BY 4.0) should be included for each asset.
- For scraped data from a particular source (e.g., website), the copyright and terms of service of that source should be provided.
- If assets are released, the license, copyright information, and terms of use in the package should be provided. For popular datasets, `paperswithcode.com/datasets` has curated licenses for some datasets. Their licensing guide can help determine the license of a dataset.
- For existing datasets that are re-packaged, both the original license and the license of the derived asset (if it has changed) should be provided.
- If this information is not available online, the authors are encouraged to reach out to the asset's creators.

13. **New assets**

Question: Are new assets introduced in the paper well documented and is the documentation provided alongside the assets?

Answer: [Yes]

Justification: The primary new asset we introduce in this paper is the GPLQ method itself, along with our plan to release an "easy-to-use open-source toolkit supporting multiple vision tasks" (as stated in the Abstract, Section 1, and Section 3.3). This paper serves as the primary documentation for the GPLQ methodology, detailing its empirical motivations, the two-stage framework (Act-QAT and Weight-PTQ), the use of PCA-based feature mimicking, and its application to various tasks. We believe the paper provides a thorough explanation of how GPLQ works and its advantages. The specifics of the toolkit's documentation will accompany its release, but the core principles and functionality are described herein.

Guidelines:

- The answer NA means that the paper does not release new assets.
- Researchers should communicate the details of the dataset/code/model as part of their submissions via structured templates. This includes details about training, license, limitations, etc.
- The paper should discuss whether and how consent was obtained from people whose asset is used.
- At submission time, remember to anonymize your assets (if applicable). You can either create an anonymized URL or include an anonymized zip file.

14. **Crowdsourcing and research with human subjects**

Question: For crowdsourcing experiments and research with human subjects, does the paper include the full text of instructions given to participants and screenshots, if applicable, as well as details about compensation (if any)?

Answer: [NA]

Justification: Our research presented in this paper does not involve any crowdsourcing experiments or direct research with human subjects. All experiments were conducted using standard, publicly available computer vision datasets (ImageNet, COCO, FGVC datasets) for training and evaluating our model quantization method, GPLQ.

Guidelines:

- The answer NA means that the paper does not involve crowdsourcing nor research with human subjects.

- Including this information in the supplemental material is fine, but if the main contribution of the paper involves human subjects, then as much detail as possible should be included in the main paper.
- According to the NeurIPS Code of Ethics, workers involved in data collection, curation, or other labor should be paid at least the minimum wage in the country of the data collector.

15. **Institutional review board (IRB) approvals or equivalent for research with human subjects**

Question: Does the paper describe potential risks incurred by study participants, whether such risks were disclosed to the subjects, and whether Institutional Review Board (IRB) approvals (or an equivalent approval/review based on the requirements of your country or institution) were obtained?

Answer: [NA]

Justification: Our research, which focuses on algorithmic development for model quantization using existing public datasets, did not involve human subjects. Consequently, considerations of potential risks to study participants, disclosure of such risks, or the necessity for Institutional Review Board (IRB) approvals (or equivalent) are not applicable to this work.

Guidelines:

- The answer NA means that the paper does not involve crowdsourcing nor research with human subjects.
- Depending on the country in which research is conducted, IRB approval (or equivalent) may be required for any human subjects research. If you obtained IRB approval, you should clearly state this in the paper.
- We recognize that the procedures for this may vary significantly between institutions and locations, and we expect authors to adhere to the NeurIPS Code of Ethics and the guidelines for their institution.
- For initial submissions, do not include any information that would break anonymity (if applicable), such as the institution conducting the review.

16. **Declaration of LLM usage**

Question: Does the paper describe the usage of LLMs if it is an important, original, or non-standard component of the core methods in this research? Note that if the LLM is used only for writing, editing, or formatting purposes and does not impact the core methodology, scientific rigorousness, or originality of the research, declaration is not required.

Answer: [NA]

Justification: The core methodology of our research in this paper, which introduces and evaluates the GPLQ quantization framework for Vision Transformers, does not involve the use of Large Language Models (LLMs) as an important, original, or non-standard component. Our work is focused on quantization techniques, empirical analysis, and algorithmic design within the computer vision domain. While we mention adapting GPLQ for LLMs as a future work direction (Section 6), LLMs were not part of the development or execution of the research presented.

Guidelines:

- The answer NA means that the core method development in this research does not involve LLMs as any important, original, or non-standard components.
- Please refer to our LLM policy (https://neurips.cc/Conferences/2025/LLM) for what should or should not be described.

