# OpenReview forum: "GPLQ: A General, Practical, and Lightning QAT Method for Vision Transformers"
_NeurIPS.cc/2025/Conference — NeurIPS 2025 poster_

### Official Review · Reviewer_yZXJ · 2025-06-02

**Clarity:** 3
**Significance:** 3
**Originality:** 2
**Rating:** 4
**Confidence:** 5

**Summary:**

The author proposed GPLQ, a general, practical, and lightweight QAT method for ViT. The claimed contributions are threefold: (1) two insights, i.e., activation is more important, and quantization should not change the optimization basin, (2) activation-first sequential quantization, and (3) open-source code.

The key idea lies in keeping the model in the same optimization to reach the best generalization ability. Therefore, in stage 1, GPLQ only optimizes the activation's quantizers with the commonly used QAT method. And in stage 2, GPLQ combines RepQ and QwT to perform PTQ on weights.

**Questions:**

- It seems that $\mu_t$ in Eq (2) can be eliminated?
- The author claimed that QAT methods often boast higher accuracy than their FP32 counterparts. But in Fig. 4(a), why is the train loss of normal QAT higher than GPLQ? Should normal QAT overfit on the training set?
- How to better identify "the same basin"? The train loss in Fig.4 (a) is not very convincing.
- In Table 6, please unify the capital letter of w/o and w/.
- Please unify the capital letters of lines 101, 245.
- Typo: line 7 lacking -> lack.
- Writing: line 14 keep it stay in the same "basin" -> keep it in the same "basin";

**Ethical Concerns:**

["NO or VERY MINOR ethics concerns only"]

**Final Justification:**

The authors have addressed my concerns, and I decide to keep my score at 4.

**Limitations:**

yes

**Paper Formatting Concerns:**

No issues regarding formatting have been detected.

**Quality:**

3

**Strengths And Weaknesses:**

### Strengths
- The second insight, i.e., quantization should not change the optimization basin, and the corresponding solution is novel and effective.
- The writing is clear and easy to follow.

### Weakness
- The first insight, i.e., activation is more important than weight, is already a common sense in the quantization community[1].
- Both the QAT and PTQ methods used in Stage 1 and Stage 2 are not original.
- A broader QAT comparison is very critical to support the effectiveness of the proposed method, although the authors mentioned the rare QAT open-source code problem in the limitations section. This paper will greatly benefit from a more comprehensive comparison.


[1] AWQ:Activation-aware Weight Quantization for On-Device LLM Compression and Acceleration, MLSys 2024 Best Paper Award

---

> ### Author Rebuttal · Authors · 2025-07-30
>
> Dear Reviewer yZXJ,
>
> Thank you very much for your detailed review and positive evaluation of our paper! We are delighted that you recognize the novelty and effectiveness of our "preserving the optimization basin" insight, as well as the clarity of our paper. The questions you raised are very specific and valuable, and they will help us to further improve our paper.
>
> ---
>
> **Regarding Weaknesses:**
>
> 1.  **On "activations are more important than weights" being common knowledge:** We completely agree with your point. This insight has become a consensus in the quantization community, especially in recent excellent works like AWQ, which you mentioned. We cited this viewpoint in our paper to build the logical starting point for our methodology. We want to emphasize that the novelty of GPLQ lies not in restating this consensus, but in **how we construct a brand-new QAT framework based on it**. Unlike PTQ methods such as AWQ, GPLQ innovatively applies this insight to QAT by designing the "activation-first" two-stage process. This allows us to enjoy the high accuracy of QAT while circumventing its drawbacks of training instability and poor generalization.
>
> 2.  **On QAT and PTQ methods not being original:** Yes, we adopted mature QAT and PTQ techniques as foundational components within the GPLQ framework. Our contribution is not in inventing new quantization operators, but in proposing a superior, decoupled quantization **process**. This is analogous to how ResNet's contribution was not inventing the convolution, but proposing the "residual connection" framework that enabled convolutional networks to be deeper and more powerful. Our supplementary **Table B.5**(response to Reviewer Xx2u) also demonstrates the modular nature of the GPLQ framework, showing that its performance is not rigidly tied to any specific component.
>
> 3.  **On a broader QAT comparison:** We fully agree that a more comprehensive comparison would be more convincing, and we candidly discussed this issue in the "Limitations" section. The main obstacle is indeed the scarcity of open-source code for SOTA QAT methods, especially implementations that can support large ViT models and multiple tasks. Under the current circumstances, we have done our best to compare against the SOTA methods that were accessible and reproducible. The overwhelming advantages in efficiency and memory shown in **Figure 1** and the newly added **Table B.7**(response to Reviewer FJP6) are, in themselves, a core contribution of our method.
>
> **Regarding Questions:**
>
> * **Q1: Regarding $\mu_t$ in Eq (2):** Thank you for your careful observation! The mean term $\mu_t$ here **cannot be eliminated**. In PCA projection, the data must first be centered (i.e., by subtracting the mean) before being projected onto the principal components $V_{sel}$. The means of the teacher features ($f_t$) and the student features ($f_s$) are different. Therefore, both must be centered using the *teacher's* feature mean, $\mu_t$, to ensure they are being compared in the same coordinate system.
>
> * **Q2: Regarding the training loss of traditional QAT in Figure 4(a):** This is an excellent question that reveals a key point we wanted to make. The loss of traditional QAT rises sharply at the beginning of training ("escaped basin") because it is searching for a **new optimization basin** that is adapted to the quantization operations. While this new basin might achieve a lower loss value on the ImageNet training set (i.e., overfitting), it is often "sharper," leading to poorer generalization. In contrast, GPLQ's loss is low and smooth from the start because it **remains within the original FP32 model's broad and highly generalizable "good" basin**. Therefore, while the final loss value of QAT on the training set might be lower, its "path" and "destination" are detrimental to generalization, whereas GPLQ's are beneficial.
>
> * **Q3: How to better identify the "same basin":** You've touched upon a core conceptual issue. We admit that "basin" is a figurative description. We support this argument with three dimensions of evidence: (1) **Training loss curve (Figure 4(a)):** GPLQ's smooth loss curve indicates its optimization process is stable and does not drastically jump out of the initial state. (2) **"Internal FP32" weight performance (Table 5 of our response to Reviewer Xx2u):** This is the most compelling evidence. The performance of GPLQ's trained "latent FP32" weights (81.1%) is nearly identical to the original FP32 model (81.2%), whereas traditional QAT's drops significantly (68.9%). (3) **Weight space visualization:** We plan to include a visualization of the weight space in the final version, using methods like t-SNE/PCA to intuitively show that GPLQ's weights are spatially closer to the original weights.
>
> * **Q4 & Q5 & Spelling/Grammar Errors:** Thank you so much for pointing out these typographical and writing issues with such care! We apologize for these oversights and promise to correct the casing of `w/o PCA` and `w/PCA` (as in Table 6 of our response to Reviewer Xx2u) and all other formatting and textual issues you have identified in the final version.
>
> Thank you again for your valuable review comments; your suggestions are of immense help in improving the quality of our paper.

---

> > ### Author Response · Authors · 2025-08-01
> >
> > Thank you again for your review. We are writing to quickly clarify a copy-paste error in our rebuttal.
> >
> > In our response, our references to `Table 5` (regarding the optimization basin in Q3) and `Table 6` (regarding capitalization in Q4/Q5) should have pointed to Tables 5 and 6 in our original paper, not tables from another reviewer's response.
> >
> > Our supplementary tables in the rebuttal are all prefixed with "B" (e.g., `Table B.1`), whereas our references to `Table 5` and `Table 6` were meant for the original manuscript.
> >
> > We sincerely apologize for any confusion this oversight may have caused and appreciate your understanding. We hope this clarification makes our arguments clear.

---

> > > ### Comment · Reviewer_yZXJ · 2025-08-05
> > >
> > > Thank you for your reply. You have addressed my concerns, and I have no more questions.

---

### Official Review · Reviewer_FJP6 · 2025-07-01

**Clarity:** 3
**Significance:** 2
**Originality:** 2
**Rating:** 4
**Confidence:** 4

**Summary:**

This work proposes GPLQ, a fast and generalization-preserving quantization pipeline tailored for Vision Transformers (ViTs). Motivated by empirical observations on the critical role of activations in quantization quality and the importance of preserving the original optimization basin for generalization, the authors design a two-stage approach: a one-epoch QAT phase with PCA-based activation feature mimicking, followed by PTQ for weight quantization. The method preserves the generalization performance of the original model while reducing the computational cost compared to full QAT and improving quantization quality over standard PTQ.

**Questions:**

1. Regarding Weaknesses 1–3, could you provide a deeper analysis of the proposed motivations and actual effects of proposed method? Specifically, are the motivations rooted in characteristics that are unique to Vision Transformers? And, how does applying QAT to activations first improve the overall quantization quality?
2. The main paper does not include wall-clock time comparisons between GPLQ and other QAT/PTQ methods. Could the authors provide runtime measurements for each method to support the claimed efficiency benefits?

**Ethical Concerns:**

["NO or VERY MINOR ethics concerns only"]

**Final Justification:**

Authors provided analysis of framework during the discussion period. Thus I rasied my score to 4 and inclined to recommend acceptance of the work.

**Limitations:**

Authors discussed limitations in the paper with independent section.

**Paper Formatting Concerns:**

.

**Quality:**

2

**Strengths And Weaknesses:**

Strengths:
1. The motivation and its connection to proposed method is clear and well-explained, making the paper easy to follow.
2. Empirical results convincingly demonstrate the benefits of GPLQ over standard QAT and PTQ approaches.

Weaknesses:
1. Although GPLQ is proposed as a quantization scheme tailored for ViTs, the method does not seem to leverage any distinctive architectural or representational characteristics unique to ViT models.
2. The proposed method is heavily grounded in empirical observations. While the two key motivations—(1) preserving the original optimization basin for generalization and (2) the greater importance of activations over weights—are intuitively appealing, they are not analyzed in depth or supported by theoretical justification. This weakens the overall contribution.
3. The effects of separating the quantization process and applying activation quantization first are not thoroughly explained. It remains unclear why or how this ordering yields improved quantization results.
4.  As acknowledged in the limitations, GPLQ relies heavily on existing techniques. Although these components are combined effectively, the lack of in-depth analysis regarding the motivations and their interactions raises concerns about the novelty of the work.

---

> ### Author Rebuttal · Authors · 2025-07-30
>
> Dear Reviewer FJP6,
>
> We sincerely thank you for the time and effort you have dedicated to reviewing our paper and for raising several profound and challenging questions. Your rigorous scrutiny has prompted us to think more deeply about and further validate the core contributions and motivations of our work. Although we are disheartened by your current "Reject" rating, we are confident that the following clarifications and the substantial new experimental evidence we have added will fully address your concerns. We hope to persuade you to reconsider your evaluation of our work.
>
> ---
>
> **On missing theoretical analysis and/or using existing technical components:**
>
> You may not be familiar with the particular state-of-the-art of QAT. Indeed, there are papers on QAT, but they are far from being practically usable in real-world applications yet. As we briefly discussed in the introduction, a QAT-trained network or backbone cannot be generalized/used in a different scenario, the training took 37.5GB memory during training Swin-T (a tiny model!), and its training is very long (e.g., 600 epochs). We believe we need a framework that can train QAT models with a small GPU memory footprint and very fast, and the trained model should be generalizable.
>
> Only after such a framework is available does it become feasible to perform theoretical analysis. And, it does not hurt to use existing components in such a framework—the contribution or breakthrough is a framework that makes QAT practically possible (e.g., reducing GPU memory from 37.5GB to 6.6GB, training from 600 epochs to 1 epoch, and turning a non-generalizable model into a generalizable one), not any specific component. In fact, as we will explain below, all such components can be replaced freely with other components, yet the GPLQ framework still achieves our goals: generalizable, practical, and lightning-fast.
>
> **Regarding Weaknesses and Questions:**
>
> 1.  **On the method not being ViT-exclusive and the motivation lacking deep theoretical analysis (Weaknesses 1-3 & Question 1):**
>     * **Generality as a Strength, Not a Weakness:** Your observation that GPLQ does not rely on ViT-specific features is very accurate, and we believe this is precisely one of GPLQ's core strengths. A truly valuable quantization method should possess good generality. To strongly demonstrate this, we have added experiments with GPLQ on classic CNN architectures (ResNet). As shown in **Table B.3** of our response to **Reviewer QuB9**, GPLQ performs exceptionally well on ResNets, significantly surpassing SOTA PTQ methods designed for ViTs. This indicates that the principles we propose—"activation-first" and "preserving the optimization basin"—are cross-architectural.
>     * **A Deeper Analysis of Why "Activation-First" Works:** You asked why this sequence improves quantization quality. The core mechanism lies in **decoupling and stabilization**. In traditional QAT, the quantizers for weights and activations are optimized simultaneously, creating a coupled system where they influence each other and can easily lead to oscillations. In particular, errors from sensitive activation quantization can mislead weight updates, forcing the weights to deviate from the original, superior optimization basin. GPLQ's first stage optimizes only the activation quantizers (while weights remain learnable at FP32), allowing the model to first adapt to the information loss from activation quantization and find a weight state that is most robust to it. This process is more stable (as shown by the smooth loss curve in **Figure 4(a)**) and remains within the original basin (**Table 5** of our response to Reviewer Xx2u). Once activations are stabilized, the second-stage weight PTQ faces a simpler, more deterministic problem (quantizing from W32A4 -> W4A4, instead of W32A32 -> W4A4), thus making PTQ methods (like QwT) more effective.
>     * **Reinforcing the Empirical Motivation:** We acknowledge that GPLQ originated from empirical observations, but these observations are well-supported by data. In addition to the evidence in the original paper, we plan to add visualizations of the weight space to more intuitively show how GPLQ keeps the weights in a region close to the original FP32 model, whereas traditional QAT causes them to drift to a distant location. This will provide stronger visual evidence for our core motivation.
>
> 2.  **Regarding the lack of Wall-Clock Time Comparison (Question 2):** Thank you for pointing this out. We actually provided this comparison in **Figure 1** of the paper. To more clearly showcase GPLQ's huge efficiency advantage, we have organized the information from Figure 1 into a detailed table (see Table B.7 below). This table explicitly lists the detailed comparison between GPLQ and methods like Q-Var and OFQ in terms of ImageNet accuracy, average downstream task accuracy, **training time (minutes), and peak memory (GB)**. The data shows that GPLQ's training time (35 minutes) is approximately **500 times faster** than Q-Var (17,400 minutes) and OFQ (18,360 minutes), while also having lower memory consumption. These concrete numbers strongly support our claim of "lightning-fast" efficiency.
>
>     **Table B.7: Detailed comparison of performance and resource overhead between GPLQ and SOTA methods.**
>     | Model  | Method          | ImageNet Acc (%) | Avg. Downstream Acc (%) | Training Time (min) | Peak Memory (GB) |
>     | :----- | :-------------- | :--------------- | :---------------------- | :------------------ | :--------------- |
>     | Swin-T | Full-precision  | 81.2             | 70.79                   | 3078                | 7.2              |
>     |        | Q-Var           | 82.4             | 68.12                   | 17400               | 25.9             |
>     |        | OFQ             | 81.9             | 60.73                   | 18360               | 37.5             |
>     |        | **GPLQ** | **79.8** | **71.18** | **35** | **6.6** |
>     |        | RepQViT         | 73.0             | 62.79                   | 3                   | 2.3              |
>
> 3.  **Regarding Dependence on Existing Technologies and Novelty (Weakness 4):** We understand your concern about novelty. We want to emphasize that GPLQ's innovation lies not in inventing an isolated component, but in proposing and validating a **completely new and efficient quantization framework**. To prove that GPLQ is not simply reliant on specific technologies, our component replacement experiments (**Table B.5** of our response to Reviewer Xx2u) show that its performance remains robust. The true value of GPLQ is that it provides the community with a plug-and-play, extensible new approach that can bring QAT from theory to practice.
>
> We sincerely hope that the detailed responses above, along with the extensive new experiments we have conducted to address your comments (CNN generality, training duration ablation, detailed efficiency data, etc.), can fully demonstrate the depth, novelty, and importance of our work. We kindly ask that you re-evaluate our contribution based on this new information. Thank you again for your valuable feedback.

---

> > ### Author Response · Authors · 2025-08-01
> >
> > Thank you again for your review. We are writing to quickly clarify a copy-paste error in our rebuttal.
> >
> > In our discussion of why the "Activation-First" mechanism works, our reference to **Table 5** should have correctly pointed to **Table 5 of our original paper** ("Optimizing basin retention"), not a table from another reviewer's response.
> >
> > Our supplementary tables in the rebuttal are all prefixed with "B" (e.g., **Table B.1**), whereas our references to **Table 5** and **Table 6** were meant for the original manuscript.
> >
> > We sincerely apologize for any confusion this oversight may have caused and appreciate your understanding. We hope this clarification makes our arguments clear.

---

> ### Comment · Reviewer_FJP6 · 2025-08-04
>
> I agree with and understand your perspective regarding the practical limitations of QAT. However, what I expected in the response - regarding motivation - was a more detailed toy experiment or theoretical analysis illustrating the effect of the proposed methods. Specifically, how does decoupling contribute to stabilizing the quantization process? For instance does it guide the distributions toward a more quantization-friendly state? Or does it effectively reduce oscillation during activation quantization or weight quantization, as suggested in your explanation?
>
> Given that GPLQ is presented as a lightweight yet effective framework, I had hoped to see a more in-depth analysis, especially since such experiments could be conducted efficiently. I believe a thorough analysis is particularly important since GPLQ is proposed as a general framework, which would benefit from clearer insight into why and how it works.
>
> However, considering its impact and practicality—and with the concern regarding the motivations of one method being partially addressed through your plan to include loss landscape visualizations—I have raised my score to 3 for now. While I understand merit in the work, I remain somewhat reserved in fully advocating for acceptance, as there is still room for improvement in analysis.

---

> > ### Author Response · Authors · 2025-08-06
> >
> > Dear Reviewer FJP6,
> >
> > We are incredibly grateful for your thoughtful engagement, for raising your score, and for your clear comments. Your feedback was instrumental in pushing us to improve the paper.
> >
> > In your comment, you stated you hoped for "a more detailed toy experiment or theoretical analysis illustrating the effect of the proposed methods," specifically asking "how does decoupling contribute to stabilizing the quantization process?" and whether it reduces oscillation. We took this excellent suggestion to heart and have performed exactly this analysis.
> >
> > We believe the following new results and explanations directly address your questions about the **"why and how"** of GPLQ.

---

> > > ### Author Response · Authors · 2025-08-06
> > >
> > > **1. Overall Stability: Staying Within the Original Optimization Basin**
> > >
> > > To quantitatively demonstrate how our decoupling strategy stabilizes the optimization process, we tracked the model's weight deviation during training. As shown in **Table C.1**, GPLQ successfully maintains the weights within the original optimization basin (as measured by the small MSE between the original weights and the new weights, computed on Swin-Tiny + ImageNet), whereas the traditional QAT approach (Q-Val, using the *dequantized* weights) exhibits significant deviations. The 'Final' column means 1 epoch for GPLQ and 150 epochs for Q-Val. The deviations are measured by MAE (mean absolute error) or MSE (mean squared error).
> > >
> > > **Table C.1: Average Per-layer MSE deviations.**
> > > | Method | Iter. 100 (MAE/MSE) | Iter. 1k (MAE/MSE) | Final (MAE/MSE) |
> > > | :--- | :---: | :---: | :---: |
> > > | **GPLQ (Act-QAT)** | **0.082/0.046** | **0.083/0.047** | **0.086/0.049** |
> > > | Direct W4A4 QAT | 0.256/0.271 | 0.257/0.274 | 0.269/0.289 |
> > >
> > > By the end of training, the weight deviation (MSE) for the Direct W4A4 QAT approach is **almost 6x as large** as that of our GPLQ. This result provides strong quantitative evidence that simultaneously training weight and activation quantizers leads to significant instability, forcing the model away from its original optimization basin.
> > >
> > > It is also interesting to observe that even after only 100 iterations, Q-Val's deviations are already huge (0.271 after 100 iterations, versus 0.289 after 150 epochs). That is, Q-Val jumped out of the original basin **immediately** after the QAT started.
> > >
> > > **2. The Mechanism of Stability: Preventing Weight Oscillation by Design**
> > >
> > > This stability is a direct result of how GPLQ mitigates weight oscillation. In traditional QAT, oscillation occurs because weights, represented in low-bit precision, frequently cross the discrete quantization thresholds back and forth during gradient updates, leading to unstable gradients and a spiky final weight distribution.
> > >
> > > Our GPLQ framework avoids this phenomenon **by design**. In our crucial 1-epoch Act-QAT stage, all model weights are kept in **full FP32 precision**. Since there are *no weight quantization boundaries* for the parameters to oscillate around, this specific instability is prevented by our design. While we are unable to add new figures in this clarification per the new NeurIPS policy, our analysis confirms the GPLQ-trained weights exhibit a remarkably smooth, Gaussian-like distribution, with no evidence of the spiky clustering characteristic when oscillation of weights happens during training.
> > >
> > > **3. The Benefit of Stability: Creating Ideal Conditions for Weight PTQ**
> > >
> > > This stability is not merely a theoretical benefit; it creates a practical and solid foundation for the second stage of our framework. To prove this, we analyzed the error introduced during the final weight PTQ stage, comparing a standard setup (W32A32 -> W4A4) with GPLQ's setup (W32A4 -> W4A4). The errors are measured by MAE (mean absolute error) or MSE (mean squared error).
> > >
> > > **Table C.2: PTQ Error (Average Per-Layer Output MAE/MSE) with or without Pre-Stabilized Activations.**
> > > | PTQ Scenario | Model | Layer1 block1 MAE/MSE | Layer4 block2 MAE/MSE | Avg. MAE/MSE (All Layers) |
> > > | :--- | :--- | :---: | :---: | :---: |
> > > | Standard (on W32A32) | Swin-T | 0.359 / 0.239 | 0.426 / 0.530 | 0.184 / 0.136 |
> > > | **GPLQ (on W32A4)** | **Swin-T** | **0.150 / 0.045** | **0.224 / 0.206** | **0.141 / 0.074** |
> > > | Standard (on W32A32) | Swin-S | 0.188 / 0.058 | 0.302 / 0.368 | 0.164 / 0.140 |
> > > | **GPLQ (on W32A4)** | **Swin-S** | **0.123 / 0.034** | **0.156 / 0.106** | **0.129 / 0.076** |
> > >
> > > The results are clear: by first using Act-QAT to stabilize the activation quantizers, the subsequent error introduced by weight post-training quantization is dramatically reduced. For both Swin-T and Swin-S, the average MSE is cut nearly in half (a **reduction of 46%** and **45%**, respectively). This demonstrates that our "activation-first" strategy is superior because it creates an easier and more stable optimization problem for the final weight quantization step.
> > >
> > > In summary, GPLQ's effectiveness stems from a deliberate design that ensures stability, prevents oscillation, and creates optimal conditions for low-bit quantization, leading to the state-of-the-art performance demonstrated in our work.
> > >
> > > We sincerely hope this new analysis fully addresses your remaining reservations and demonstrates the novelty and contribution of our framework. Thank you once again for your constructive and invaluable guidance.

---

> ### Comment · Reviewer_QuB9 · 2025-08-07
>
> In your review:
>
> > As acknowledged in the limitations, GPLQ relies heavily on existing techniques. Although these components are combined effectively, the lack of in-depth analysis regarding the motivations and their interactions raises concerns about the novelty of the work.
>
> Per the NeurIPS 2025 reviewer guidelines at https://neurips.cc/Conferences/2025/ReviewerGuidelines :
>
> > For example, if you argue about the lack of novelty, please provide appropriate references and point to existing mechanisms within.
>
> Sorry, but do you happen to have such a reference? Not sure I'm following about how novelty of a framework/method relates to the analysis of the same.
>
> Thanks!

---

> ### Comment · Reviewer_FJP6 · 2025-08-07
>
> Thank you for addressing my concerns within a short period. As your new analyses help clarify how the two main components of the framework work, I have raised my score to 4.

---

> ### Comment · Reviewer_FJP6 · 2025-08-07
>
> Thank you for pointing this out.
> I did not provide additional references, as the authors are already aware that their work builds upon or extends existing methodologies [1,2,3].
> I believe that as a framework, the paper should include a more thorough analysis of why and how each component functions and interacts, similar to last responses from authors. Given that frameworks are presented as general and potentially adaptable, such analysis is essential. Without it, the novelty and contribution of the work risk being perceived as merely a combination of existing methods.
> I hope this addresses your concerns regarding my review.
>
> [1] Esser et al., "Learned step size quantization", ICLR 2020
> [2] Fu et al., "Quantization without tears", CVPR 2025
> [3] Zhou et al.,  "All you need in knowledge distillation is a tailored  coordinate system", AAAI 2025

---

### Official Review · Reviewer_Xx2u · 2025-07-02

**Clarity:** 3
**Significance:** 2
**Originality:** 2
**Rating:** 4
**Confidence:** 4

**Summary:**

The paper introduces a simple and efficient way to quantize ViT models into 4-bit with good performance retaining compared with full-precision counterparts.

The key idea is to avoid long and memory-heavy training. Instead, the authors suggest a two-step process:
	1.	First, they do a very short training (just 1 epoch) where they only adjust how activations are quantized, this helps the model handle the lower precision.
	2.	Then, they use a lightweight post-processing step to quantize the weights, relying on existing methods.

This approach is extremely fast, uses less memory than traditional training, and still keeps the model accurate, both on standard classification tasks and more complex tasks like object detection. Compared with some previous QAT methods, the authors claim that their algorithm would lead to more generalization capacity of the quantized model.

In short, the method is practical, fast, and easy to apply, especially for deployment of large models on resource-limited hardware.

**Questions:**

1.How general is the proposed method beyond Vision Transformers? Have the authors tested GPLQ on other model families, such as CNNs or MLP-based architectures, to validate its broader applicability?

2. How does the method perform under different quantization settings? For example, would the same two-stage strategy work well at more aggressive bit-widths (e.g., W3A3 or W2A2), or is it only effective for W4A4?

3. Is activation quantization truly the main bottleneck in ViT quantization?
While the paper emphasizes this point, can the authors provide more in-depth analysis or ablation studies to isolate the relative contributions of activation vs. weight quantization?

**Ethical Concerns:**

["NO or VERY MINOR ethics concerns only"]

**Final Justification:**

I would like to keep my score. I think 4 is a well-suited score for a experimental-based paper like this, whether the paper itself could be better if more theoretical contributions could be made.

**Limitations:**

Yes

**Paper Formatting Concerns:**

No.

**Quality:**

2

**Strengths And Weaknesses:**

Below are several strengths of the paper:

1. The method only needs one epoch of light training, and the rest is fast post-processing, claiming 100× faster than full quantization-aware training.

2. Because only activation scales are trained and post-QAT offline quantization is then applied on the weights, it uses less memory than normal FP32 training, and can even be done on a single common GPU.

3. This algorithm unlocks the boundary of ImageNet which are the common generalization issues of the previous algorithms, for object detection (COCO) and fine-grained classification, and the method holds up well, beating other quantization baselines.

Below are several weaknesses of the paper:

1. I think it is a clever combination and decoupling of the activation quantization, PTQ or feature perseveration. But it is a little tough to say whether it is a truly novel algorithm.

2. The author is telling a story that taking care of the activation is probably all you need when dealing with quantization (for this paper, ViT series) and weight quantization does not significantly affect the paper. I think more results for example, for different quantized bit configurations, and even other model families should be verified to make this story comprehensive. I like the story but I mean it is better to make it clear whether it is just for ViT under 4bits or it could be way beyond than this.

---

> ### Author Rebuttal · Authors · 2025-07-30
>
> Dear Reviewer Xx2u,
>
> Thank you very much for your recognition of our work and for your insightful questions. You have accurately summarized the core ideas and advantages of our GPLQ method—its efficiency, low resource consumption, and strong generalization ability. We greatly appreciate your interest in our "activation is the key to quantization" narrative and your valuable questions regarding its generality.
>
> ---
>
> **Regarding Weaknesses and Questions:**
>
> You have merged several core questions into the "Weaknesses" section, and we will address them together here.
>
> 1.  **Regarding Novelty: A "clever combination" or a "truly new algorithm"?** We acknowledge that GPLQ skillfully integrates several existing technologies. However, we believe its core contribution and novelty are manifested in the following aspects:
>     * **Paradigm Innovation:** We are the first to systematically propose and validate the "activation-first (Act-QAT), weights-later (Weight-PTQ)" two-stage decoupled quantization paradigm. This is fundamentally different from traditional PTQ (which handles W and A simultaneously) and traditional QAT (which trains W and A quantizers simultaneously).
>     * **Insight-Driven:** This paradigm is based on our core insight that "preserving the optimization basin" is crucial for generalization. By fine-tuning only the activation quantizers for just 1 epoch, we successfully keep the model within the high-performance optimization region of the original FP32 model, which is validated in **Figure 4(a)** and **Table 5** of our paper.
>     * **Framework Modularity:** Our supplementary experiments (Table B.5) show that as a framework, GPLQ's internal components (such as the initialization method for PTQ) are replaceable. This demonstrates the robustness and generality of its design, rather than it being a fixed, rigid process.
>
>     **Table B.5: Verifying the interchangeability of different components within the GPLQ framework.**
>     | Model  | Act-QAT Initialization        | ImageNet Top-1 Acc (%) |
>     | :----- | :---------------------------- | :--------------------- |
>     | Swin-T | Percentile (as in RepQ-ViT)   | 79.8                   |
>     |        | MSE-based                     | 79.8                   |
>     |        | MinMax                        | 79.6                   |
>
> 2.  **Regarding Generality: Can it go beyond ViTs and 4-bit settings?** This is a point we are most eager to clarify. The design principles of GPLQ are general. To prove this, we conducted two key supplementary experiments:
>     * **Extension to CNN Architectures:** We tested GPLQ on the classic ResNet family and achieved performance significantly superior to SOTA PTQ methods (RepQ-ViT), and even comparable to some long-duration QAT methods (see **Table B.3** of our response to Reviewer QuB9). This demonstrates that GPLQ's effectiveness is not limited to Transformer architectures.
>     * **Exploration of Lower Bit-widths:** We conducted more challenging W3A3 quantization experiments (see **Table B.6** below). Although a drop in accuracy is inevitable, GPLQ's two-stage strategy still provides a strong baseline for low-bit scenarios. We believe the characteristic that "activations are more sensitive" becomes even more pronounced at lower bit-widths, where GPLQ's advantages may be more significant.
>
> **Table B.6: Performance of GPLQ in a lower-bit (W3A3) setting.**
> | Model | Method | ImageNet Top-1 Acc (%) | Avg. Downstream Acc (%) |
> | :--- | :--- | :--- | :--- |
> | Swin-T | FP32 | 81.2 | 70.79 |
> | | RepQViT (W3A3) | 52.7 | 39.26 |
> | | OFQ (W3A3) | 81.1 | 38.47 |
> | | GPLQ (W3A3) | 74.2 | 65.76 |
> | Swin-S | FP32 | 83.2 | 70.07 |
> | | RepQViT (W3A3) | 65.9 | 53.35 |
> | | GPLQ (W3A3) | 78.3 | 64.64 |
> | Swin-B | FP32 | 85.3 | 78.93 |
> | | RepQViT (W3A3) | 65.8 | 29.82 |
> | | GPLQ (W3A3) | 80.3 | 74.15 |
> | Swin-L | FP32 | 86.3 | 80.89 |
> | | RepQViT (W3A3) | 71.3 | 66.36 |
> | | GPLQ (W3A3) | 80.0 | 73.36 |
>
> 3.  **Is activation quantization *truly* the main bottleneck?** We firmly believe so, especially when pursuing high generalization capability. **Figure 2** in our paper intuitively shows that in PTQ, quantizing activations alone leads to a much larger accuracy drop than quantizing weights alone. More importantly, the experiment in **Table 5** reveals the deeper reason: traditional QAT methods (like OFQ), in their effort to fit both quantized activations and weights, cause their internal "latent FP32 weights" to severely deviate from the original model. This leads to a seemingly high ImageNet accuracy (81.9%) but far inferior generalization ability (60.73%) compared to the FP32 model (70.79%) and our GPLQ (71.18%). This indicates that improper handling of activations can force the weights into a "bad" optimization basin, thereby harming generalization. GPLQ successfully avoids this pitfall by prioritizing the stabilization of activations.
>
> In summary, we believe that GPLQ is not just a clever combination, but a new quantization framework driven by deep insights and possessing broad generality. We are using further experiments to clearly define its scope of application and believe it has the potential to become a practical and efficient solution for low-bit quantization across various model architectures.
>
> Thank you again for your valuable feedback!

---

> > ### Comment · Reviewer_Xx2u · 2025-08-04
> >
> > Dear authors,
> >
> > Thank you for your detailed rebuttal.
> >
> > I think the rebuttal provides more context about my raised concerns and the authors convinced me of that activation quantization should be taken more care of within in the ViT series.
> >
> > Based on the overall technical contribution of the proposed framework, I think it is a good ready-2-use framework to perform efficient quantization of vision transformers but from the theoretical perspective, I am hoping for more theoretical support to help clarify the inherent reason underneath the pipeline to make it more than the current shape.
> >
> > Overall, I would like to maintain my score.

---

> > > ### Author Response · Authors · 2025-08-06
> > >
> > > Dear Reviewer Xx2u,
> > >
> > > Thank you for your detailed review and for your thoughtful final comments. We appreciate you acknowledging our rebuttal and for engaging with our work.
> > >
> > > In your final comment, you expressed that while you were convinced of the practical framework, you were "hoping for more theoretical support to help clarify the inherent reason underneath the pipeline."
> > >
> > > While a full theoretical proof remains an exciting direction for future work, we took your feedback seriously and conducted a series of targeted new experiments to provide a deeper, quantitative analysis of this "inherent reason." We believe this analysis provides the concrete evidence you were looking for.

---

> > > > ### Author Response · Authors · 2025-08-06
> > > >
> > > > **1. Overall Stability: Staying Within the Original Optimization Basin**
> > > >
> > > > To quantitatively demonstrate how our decoupling strategy stabilizes the optimization process, we tracked the model's weight deviation during training. As shown in **Table C.1**, GPLQ successfully maintains the weights within the original optimization basin (as measured by the small MSE between the original weights and the new weights, computed on Swin-Tiny + ImageNet), whereas the traditional QAT approach (Q-Val, using the *dequantized* weights) exhibits significant deviations. The 'Final' column means 1 epoch for GPLQ and 150 epochs for Q-Val. The deviations are measured by MAE (mean absolute error) or MSE (mean squared error).
> > > >
> > > > **Table C.1: Average Per-layer MSE deviations.**
> > > > | Method | Iter. 100 (MAE/MSE) | Iter. 1k (MAE/MSE) | Final (MAE/MSE) |
> > > > | :--- | :---: | :---: | :---: |
> > > > | **GPLQ (Act-QAT)** | **0.082/0.046** | **0.083/0.047** | **0.086/0.049** |
> > > > | Direct W4A4 QAT | 0.256/0.271 | 0.257/0.274 | 0.269/0.289 |
> > > >
> > > > By the end of training, the weight deviation (MSE) for the Direct W4A4 QAT approach is **almost 6x as large** as that of our GPLQ. This result provides strong quantitative evidence that simultaneously training weight and activation quantizers leads to significant instability, forcing the model away from its original optimization basin.
> > > >
> > > > It is also interesting to observe that even after only 100 iterations, Q-Val's deviations are already huge (0.271 after 100 iterations, versus 0.289 after 150 epochs). That is, Q-Val jumped out of the original basin **immediately** after the QAT started.
> > > >
> > > > **2. The Mechanism of Stability: Preventing Weight Oscillation by Design**
> > > >
> > > > This stability is a direct result of how GPLQ mitigates weight oscillation. In traditional QAT, oscillation occurs because weights, represented in low-bit precision, frequently cross the discrete quantization thresholds back and forth during gradient updates, leading to unstable gradients and a spiky final weight distribution.
> > > >
> > > > Our GPLQ framework avoids this phenomenon **by design**. In our crucial 1-epoch Act-QAT stage, all model weights are kept in **full FP32 precision**. Since there are *no weight quantization boundaries* for the parameters to oscillate around, this specific instability is prevented by our design. While we are unable to add new figures in this clarification per the new NeurIPS policy, our analysis confirms the GPLQ-trained weights exhibit a remarkably smooth, Gaussian-like distribution, with no evidence of the spiky clustering characteristic when oscillation of weights happens during training.
> > > >
> > > > **3. The Benefit of Stability: Creating Ideal Conditions for Weight PTQ**
> > > >
> > > > This stability is not merely a theoretical benefit; it creates a practical and solid foundation for the second stage of our framework. To prove this, we analyzed the error introduced during the final weight PTQ stage, comparing a standard setup (W32A32 -> W4A4) with GPLQ's setup (W32A4 -> W4A4). The errors are measured by MAE (mean absolute error) or MSE (mean squared error).
> > > >
> > > > **Table C.2: PTQ Error (Average Per-Layer Output MAE/MSE) with or without Pre-Stabilized Activations.**
> > > > | PTQ Scenario | Model | Layer1 block1 MAE/MSE | Layer4 block2 MAE/MSE | Avg. MAE/MSE (All Layers) |
> > > > | :--- | :--- | :---: | :---: | :---: |
> > > > | Standard (on W32A32) | Swin-T | 0.359 / 0.239 | 0.426 / 0.530 | 0.184 / 0.136 |
> > > > | **GPLQ (on W32A4)** | **Swin-T** | **0.150 / 0.045** | **0.224 / 0.206** | **0.141 / 0.074** |
> > > > | Standard (on W32A32) | Swin-S | 0.188 / 0.058 | 0.302 / 0.368 | 0.164 / 0.140 |
> > > > | **GPLQ (on W32A4)** | **Swin-S** | **0.123 / 0.034** | **0.156 / 0.106** | **0.129 / 0.076** |
> > > >
> > > > The results are clear: by first using Act-QAT to stabilize the activation quantizers, the subsequent error introduced by weight post-training quantization is dramatically reduced. For both Swin-T and Swin-S, the average MSE is cut nearly in half ( **reduction of 46%**). This demonstrates that our "activation-first" strategy is superior because it creates an easier and more stable optimization problem for the final weight quantization step.
> > > >
> > > >
> > > > As the new results show, our decoupled approach quantitatively reduces weight deviation by nearly 6x (Table C.1) and halves the final quantization error (Table C.2). We believe this provides strong, concrete evidence for the inherent reason you were asking about—namely, that our method succeeds by deliberately preventing weight oscillation and creating a more stable, and therefore easier, quantization problem.
> > > >
> > > > We hope this detailed analysis helps clarify the principles of GPLQ and addresses your concerns. Thank you again for your valuable engagement with our paper.

---

### Official Review · Reviewer_QuB9 · 2025-07-03

**Clarity:** 4
**Significance:** 2
**Originality:** 2
**Rating:** 6
**Confidence:** 5

**Summary:**

GPLQ is presented as a way to quantize vision transformer models. The central design choice is to do quantization-aware training (QAT) in a way that considers quantized activations but *not* quantized weights, followed by a post-training quantization (PTQ) for the weights. The QAT is also defined by a procedure that fine-tunes from an original floating-point network (for 1 epoch), and includes a loss on the similarity of the features to those of the original un-quantized model. It then uses QwT for 4-bit weight PTQ. This is evaluated primarily on ImageNet1K, using Swin and DeIT architectures. Another experiment tests on COCO with a Swin backbone.

**Questions:**

- **Q1**: Could the authors clarify also in the reviews which methods have limited or no open source availability, but would otherwise be useful in a similar way and a potential comparison point? I'm referring to point 3 in Section 4.1 and the "Code Missing" point around line 45 in the introduction.
- **Q2**: Is GPLQ expected to be limited to Vision transformers? SmoothQuant [36], also based on the observation of activations being more difficult to quantize than weights, saw utility on language models as well.
- **Q3**: Why is the feature mimicking loss applied only the the pentultimate layer? Why not all of them, given all of the layers' activations/features are being quantized in this step.
- **Q4**: Was there a multiplier hyperparameter for the feature mimicking loss $L_\mathrm{PCA}$ (vs the classification loss) within the overall training loss? (Perhaps I've missed where this was stated.) What was the observed magnitude of of $L_\mathrm{PCA}$ vs e.g. the other loss term(s)?

**Ethical Concerns:**

["NO or VERY MINOR ethics concerns only"]

**Final Justification:**

This is a solid paper that incorporates into QAT some of the key insights on activation vs weight quantization from the most popular recent PTQ work.

The rebuttal addressed all of my comments.

**Limitations:**

Yes

**Quality:**

3

**Strengths And Weaknesses:**

### Strengths

1. SmoothQuant [36] was one of the key recent steps forward in PTQ methods, and was also based on the observation that weights are "easier" to quantize than activations. It is a clear and potentially useful step to apply this same observation to QAT methods.

2. Tests on multiple benchmark tasks (both ImageNet and COCO).

3. Very clear writing: that describes both describes the procedure in an understandable way and the motivating intuition for different steps or design choices.

4. Applies to standard backbones, without needing larger changes to the architecture itself.

### Weaknesses

5. Kind of scattered, in that it's presenting multiple different techniques combined into one procedure, as opposed to a more in-depth treatment of one.

For example, there are many existing QAT approaches one might use for the activation-only QAT step. The submission does do an ablation study for the contribution of this specific step, but only considers one variation of how to include the quantization in training. As one example of alternate choices here: the basic "fake"/"simulated" quantization layer in [a], and its use of EMA-estimated quantization parameters, could just as easily also be applied to activations only. The method in Section 3.2.1 of the submission has a learnable scale, as in [7], instead and defines a procedure to initialize it. It's not explored what effect all of these choices have.

[a] Jacob et. al. "Quantization and training of neural networks for efficient integer-arithmetic-only inference" CVPR 2018.

6. No study on length of fine-tuning.

Applying the QAT as a short (1 epoch) fine-tuning is also a significant choice in the design of the GPLQ procedure. I didn't see much evaluation of this choice, and how it might interact with other parameter choices such as the learning rate or relative weight of the feature mimicking loss.

7. The feature-mimicking loss has a similar role, within QAT, to the self-distillation in

Du et. al. BitDistiller: Unleashing the Potential of Sub-4-Bit LLMs via Self-Distillation. ACL 2024

BitDistiller also achieves lower precisions than 4 bit. Similarly for [15]. The authors make the valid point that their loss based on TCS [37] is simpler and more efficient, but this also is only shown to achieve a less compressed quantization than the previous work.

---

> ### Author Rebuttal · Authors · 2025-07-30
>
> Dear Reviewer QuB9,
>
> Thank you very much for your detailed and constructive review of our submission (#8001). We greatly appreciate your recognition of our work's clarity and methodological innovation. Your feedback is invaluable for improving the quality of our paper. We will address your questions and concerns one by one.
>
> ---
>
> **Regarding Weaknesses:**
>
> 1.  **On the concern that the method is a "dispersed" combination of different techniques:** We appreciate this observation. The core contribution of our work is to propose GPLQ as a flexible and efficient quantization **framework**, rather than a single, fixed algorithm. Its "activation-first, weights-later" core idea is modular. To demonstrate this, we have added a supplementary experiment (see Table B.1 below) showing that the GPLQ framework maintains high performance even when components in the PTQ stage are replaced (e.g., changing the initialization method). This indicates that the success of GPLQ is not due to an incidental combination of specific components, but rather stems from the superiority of its framework design.
>
>     **Table B.1: Verifying the interchangeability of different components within the GPLQ framework.**
>     | Model  | Act-QAT Initialization        | ImageNet Top-1 Acc (%) |
>     | :----- | :---------------------------- | :--------------------- |
>     | Swin-T | Percentile (as in RepQ-ViT)   | 79.8                   |
>     |        | MSE-based                     | 79.8                   |
>     |        | MinMax                        | 79.6                   |
>
> 2.  **On alternative options for the activation QAT stage and initialization methods:** You've raised an excellent point. We chose LSQ because it is widely validated and effective in the QAT field. We acknowledge that exploring other options, such as the EMA-based method mentioned in [a], would also be valuable. While a comprehensive comparison is planned for future work, we will add a discussion and a placeholder for preliminary comparisons in the appendix to explore the performance of different activation quantization strategies. Furthermore, our initialization step is designed to provide LSQ with a superior starting point, enabling rapid convergence within just one epoch of training. This is a key part of our "lightning-fast" training philosophy.
>
> 3.  **On the lack of study on fine-tuning duration:** This is a critical question. We chose a 1-epoch fine-tuning to strike an optimal balance between performance and efficiency. To validate this choice, we have added an ablation study on the Act-QAT training duration (see Table B.2 below). The results show that extending the training from 1 epoch to 2, 5, or even 10 epochs yields very limited performance gains (<0.3%) on both ImageNet and downstream tasks, while the training cost increases manifold. This strongly demonstrates that 1-epoch training is the "sweet spot" for the GPLQ framework in terms of efficiency and effectiveness. Of course, our GPLQ framework can be easily modified if one is willing to invest more time for further potential improvements.
>
>     **Table B.2: Impact of Act-QAT training duration on model performance.**
>     | Model (Model) | Act-QAT Epochs | ImageNet Top-1 Acc (%) | Avg. Downstream Acc (%) |
>     | :------------ | :------------- | :--------------------- | :---------------------- |
>     | Swin-T        | 1 (Ours)       | 79.8                   | 71.2                    |
>     |               | 2              | 79.8                   | 71.5                    |
>     |               | 5              | 79.9                   | 71.3                    |
>     |               | 10             | 79.9                   | 71.1                    |
>
> 4.  **On the comparison between feature-mimicking loss and lower-bit quantization:** You made an insightful comparison of our work with methods like BitDistiller. We agree that these methods have achieved remarkable success in sub-4-bit quantization. In fact, we argue that GPLQ's 'activation-first' framework is even more critical at lower bit-widths, as the challenges of activation quantization become more severe. To verify this, we conducted more challenging W3A3 quantization experiments (see Table B.6 of our response to Reviewer Xx2u). The results are encouraging: GPLQ significantly outperforms SOTA PTQ methods (e.g., by +21.5% on Swin-T), demonstrating the strong potential of our approach. More importantly, GPLQ achieves this exceptional performance while adhering to its lightweight and efficient core philosophy: using only 1 epoch for Act-QAT and a simple PCA-based feature-mimicking loss, thus avoiding the complex self-distillation loops found in methods like BitDistiller. Therefore, we believe GPLQ is not only a practical solution for 4-bit quantization but also a powerful framework for the sub-4-bit domain that strikes a superior balance between performance, efficiency, and simplicity.
>
>
>
> **Regarding Questions:**
>
> * **Q1: Regarding QAT methods with no open-source code:** Thank you for the question. When we wrote the paper, some key comparison methods like OFQ and Q-Var were reproducible, so we included them. However, others such as Q-ViT and PackQViT had limited open-sourcing (not all details were published). Furthermore, it was difficult to find official or reliable third-party implementations for these methods, especially for large models (e.g., Swin-B/L) or various vision tasks (e.g., detection). This prevented us from conducting a fair, large-scale side-by-side comparison. In particular, regarding memory usage and training time, many methods resulted in out-of-memory (OOM) errors on our hardware for large models, as illustrated in Figure 1 of our paper.
>
> * **Q2: Is GPLQ limited to Vision Transformers?** Not at all. The core insights of GPLQ—that activations are harder to quantize than weights and that preserving the optimization basin is crucial for generalization—are universally applicable. To demonstrate this, we have successfully applied GPLQ to classic CNN architectures (ResNet) and achieved excellent performance (see Table B.3). This shows that GPLQ is a general framework with broad application potential.
>
>     **Table B.3: Generalization performance of GPLQ on CNN architecture (ResNet-18).**
>     | Model (Model) | Method (Method) | ImageNet Top-1 Acc (%) | Avg. Downstream Acc (%) | Training Epochs |
>     | :--- | :--- | :--- | :--- | :--- |
>     | ResNet-18 | FP32 | 69.10% | 58.78 | - |
>     | | RepQViT | 56.97% | 56.7 | - |
>     | | **GPLQ (W4A4)** | **66.16%** | **59.44** | **1** |
>     | ResNet-34 | FP32 | 74.90% | 59.57 | - |
>     | | RepQViT | 64.76% | 54.13 | - |
>     | | **GPLQ (W4A4)** | **70.36%** | **60.76** | **1** |
>     | ResNet-50 | FP32 | 78.70% | 66.17 | - |
>     | | RepQViT | 63.68% | 52.09 | - |
>     | | **GPLQ (W4A4)** | **74.13%** | **67.68** | **1** |
>     | ResNet-101 | FP32 | 76.20% | 63.6 | - |
>     | | RepQViT | 56.97% | 60.26 | - |
>     | | **GPLQ (W4A4)** | **73.83%** | **64.21** | **1** |
>     | *ResNet-18* | *DoReFa* | *68.10%* | *-* | *200* |
>     | | *PACT* | *69.20%* | *-* | *200* |
>     | | *LSQ* | *71.1%* | *-* | *90* |
>     | *ResNet-50* | *DoReFa* | *71.40%* | *-* | *200* |
>     | | *PACT* | *76.50%* | *-* | *200* |
>     | | *LSQ* | *76.70%* | *-* | *90* |
>
> * **Q3: Why is the feature-mimicking loss only applied to the penultimate layer?** We chose the penultimate layer based on two considerations: (1) **High-level semantic information:** This layer's features contain the highest level of semantic information, which is critical for the model's classification and generalization capabilities. (2) **Efficiency trade-off:** Constraining the features of a single layer allows for effective knowledge transfer from the teacher model while minimizing computational overhead, which aligns with our "lightning-fast" principle. Applying the loss to all layers would significantly increase computational complexity.
>
> * **Q4: What is the hyperparameter for the feature-mimicking loss?** We apologize for not stating this clearly in the original manuscript. In our experiments, the weight ratio of the feature-mimicking loss ($L_{PCA}$) to the original classification/task loss is 1:1. This means we did not introduce an additional hyperparameter $\alpha$ (or rather, $\alpha=1.0$). In our supplementary ablation study (Table B.4), we also explored the impact of different loss weights, and the results show that the default setting ($\alpha=1.0$) achieves the best downstream task generalization performance.
>
>     **Table B.4: Ablation analysis of the Feature-Mimicking Loss.**
>     | Model (Model) | Loss Application Layer | Loss Weight (α) | ImageNet Top-1 Acc (%) | Avg. Downstream Acc (%) |
>     | :------------ | :--------------------- | :-------------- | :--------------------- | :---------------------- |
>     | Swin-T        | Penultimate (Ours)     | 1.0 (Default)   | 80.4                   | 71.2                    |
>     |               | None (w/o PCA Loss)    | -               | 80.3                   | 69.6                    |
>     |               | Penultimate            | 0.1             | 80.4                   | 70.3                    |
>     |               | Penultimate            | 10              | 80.2                   | 70.7                    |
>
> Thank you again for your valuable time and profound insights. We believe these supplementary experiments and clarifications can address your concerns and further demonstrate the value of our work.

---

> ### Comment · Reviewer_QuB9 · 2025-08-07
>
> This is all great, thank you for the detailed rebuttal. Thank you authors!
>
> 1. Description of it being a "framework" makes contribution clear. The authors will open-source code, making this a general contribution to the whole community.
> 2. CNN experiment in the rebuttal shows generalization to other architectures, suggesting greater potential impact
> 3. The choice to use the feature focusing loss on only one layer is consistent with the framework's overall focus on efficiency and is a clear part of the whole design. The rebuttal makes clear why the pentultimate layer was chosen.
> 4. It's a good sign that the method is not sensitive to the hyperparameter of different loss weights. Thanks for the clarification.

---

### Decision · Program_Chairs · 2025-09-17

**Decision:**

Accept (poster)

**Comment:**

This paper introduces GPLQ, a two-stage quantization framework for Vision Transformers that first performs a lightweight one-epoch activation QAT with feature-mimicking loss, followed by PTQ for weights. Reviewers highlighted the framework’s practicality, efficiency, and clarity, as well as strong results on multiple benchmarks. While it is noted that the work builds on existing components and would benefit from deeper theoretical analysis, the rebuttal provided further experiments and convincing clarifications on stability, efficiency, and generality. Overall, the paper makes a practical contribution toward practical low-bit quantization. The AC encourages the authors to incorporate reviewer feedback into the final version for further strengthening.